# The cichlid oral and pharyngeal jaws are evolutionarily and genetically coupled

Andrew J. Conith [1✉] & R. Craig Albertson [1✉]

Evolutionary constraints may significantly bias phenotypic change, while "breaking" from such constraints can lead to expanded ecological opportunity. Ray-finned fishes have broken functional constraints by developing two jaws (oral-pharyngeal), decoupling prey capture (oral jaw) from processing (pharyngeal jaw). It is hypothesized that the oral and pharyngeal jaws represent independent evolutionary modules and this facilitated diversification in feeding architectures. Here we test this hypothesis in African cichlids. Contrary to our expectation, we find integration between jaws at multiple evolutionary levels. Next, we document integration at the genetic level, and identify a candidate gene, *smad7*, within a pleiotropic locus for oral and pharyngeal jaw shape that exhibits correlated expression between the two tissues. Collectively, our data show that African cichlid evolutionary success has occurred within the context of a coupled jaw system, an attribute that may be driving adaptive evolution in this iconic group by facilitating rapid shifts between foraging habitats, providing an advantage in a stochastic environment such as the East African Rift-Valley.

[1] Biology Department, University of Massachusetts Amherst, Amherst, MA 01003, USA. ✉email: ajconith@bio.umass.edu; albertson@bio.umass.edu

The potential phenotypes available to a population are not limitless, rather, their bounds are shaped by various genetic, developmental, environmental, and functional constraints[1,2]. Constraints that act upon the phenotype can impact certain evolutionary metrics such as phenotypic disparity, and rates of taxonomic and morphological evolution, that may ultimately determine the "success" of a given clade[3–5]. Breaking from constraints involves natural selection navigating a very specific mutational landscape, but in doing so, may improve functional efficiency, or enable a population to access unoccupied ecological niche space, ultimately fostering subsequent diversification[6]. Therefore, constraints, when broken, can lead to rapid evolutionary change in a clade and may be responsible for the unevenness in rates of taxonomic and morphological evolution observed across metazoans, providing an opportunity to explore incipient stages of diversification[7,8].

Evolutionary constraints are ubiquitous across organisms and can influence the degree of variation in a trait, or the strength of covariation among traits[9,10]. Weak evolutionary constraints acting among traits could lead to many trait possibilities as each trait varies independently becoming more modular, while strong constraints may lead to fewer possibilities as traits become integrated (Fig. 1a). But how a population moves in a morphospace, with ranges defined by the strength of various constraints, is ultimately determined by the direction and magnitude of selection. Indeed, selection for shared functions or genetic pleiotropy can produce strong integration among traits, biasing phenotypic evolution to occur along a narrow line delimited by the covariation among traits––the line of least resistance[11,12]. When an evolutionary constraint is broken, some populations may deviate from this line as selection acts upon each trait independently (Fig. 1b), allowing each trait to become more specialized for a given task and permitting more unique trait combinations.

An iconic example of this process is illustrated by the feeding system in ray-finned fishes, whereby a second set of "pharyngeal" jaws have evolved by lining the posterior ceratobranchial bones of the gill arch with teeth, permitting the separation of prey capture and prey processing mechanisms (Fig. 1c-d). Functionally decoupling the jaw systems releases ray-finned fishes from mechanical constraints typically imposed by force-velocity trade-offs in single jaw systems[13]; a single jaw cannot be optimized for both power and speed simultaneously, but a two jaw complex could optimize each jaw independently for one of power or speed[14]. Within the ray-finned fishes, some clades of percomorph fishes (i.e., Cichlidae, Labridae, Pomacentridae, and Embiotocidae), have independently evolved further specializations to their pharyngeal jaws that increased feeding efficiency, specializations that are theorized to be key to their success[15].

Cichlids are characterized by rapid speciation within a diverse array of ecological niches throughout the subtropics (South-Central America, Africa, India), and the lakes of the East African Rift Valley have fostered multiple adaptive radiations, particularly within lakes Malawi, Tanganyika, and Victoria[16–18]. Cichlids owe much of their success to a highly evolvable feeding system, and in particular, the specialized pharyngeal jaw apparatus that is thought to represent a "key innovation." Greater pharyngeal jaw efficiency arises from the evolution of two synovial joints between the upper pharyngeal jaw and basicranium, fused fifth ceratobranchials, and a direct muscular connection between the neurocranium and the lower pharyngeal jaw, which together permit a stronger bite force and is predicted to provide an advantage over interspecific competitors[19]. Liem initially formulated the functional decoupling hypothesis after seminal work describing the cranial anatomy and performing kinematics of cichlid feeding[20,21]. Following Liem's work, future studies hypothesized that, alongside functional modularity, the cichlid oral and pharyngeal jaws reflect separate genetic and developmental modules, permitting variation to accumulate in each pair of jaws independently and promoting trophic diversification[22,23].

Documenting the degree of integration between oral and pharyngeal jaws across genetic, molecular, and developmental levels is crucial to formulating a more comprehensive

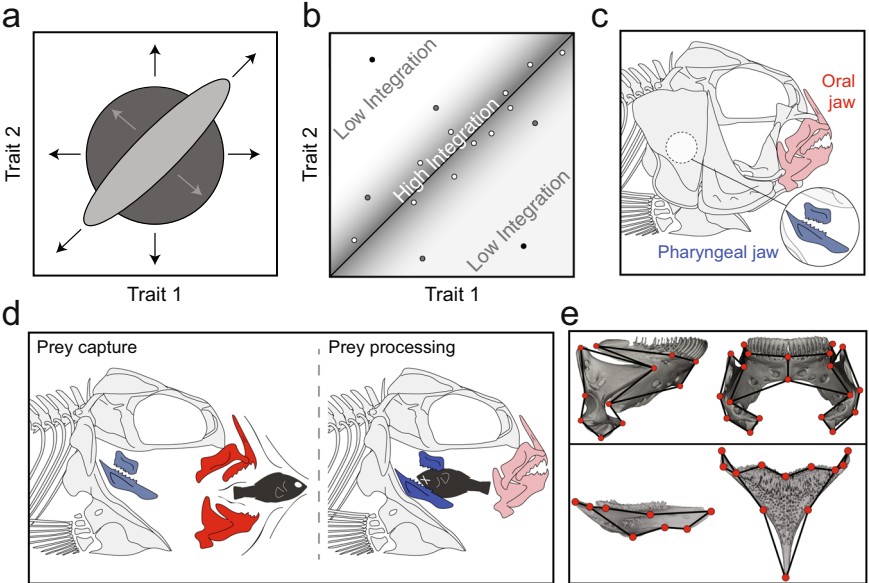

**Fig. 1 The impacts of constraints on phenotypic evolution. a** Dark gray, no constraints permitting phenotypic evolution to occur in any direction. Light gray, some constraints operating on trait evolution permitting selection to dive phenotypic change in some directions, but limiting change in others.
**b** Assessing the degree of integration between traits would reveal the strength of covariation among taxa/individuals. **c** Schematic illustrating the relative positions of the oral (red) and pharyngeal (blue) jaws within the skulls of cichlid fishes. **d** Schematic illustrating the relative roles of each jaw complex in prey capture and prey processing. **e** Oral and pharyngeal jaw symmetric landmark configurations. Top, lower oral jaw μCT scans (lateral, anterior view). Bottom, lower pharyngeal jaw μCT scans (lateral, dorsal view). Full landmark configuration information can be found in the supplement. Scan images and line drawings were produced by the authors using MeshLab (v2019-12) and Adobe Illustrator CC (v22.0.1).

understanding of what constraints have shaped cichlid evolutionary dynamics and permitted such explosive adaptive radiations. While there is some evidence to suggest the jaw systems are evolutionarily decoupled based on certain functional metrics[22,24], other work has demonstrated a degree of evolutionary and genetic integration between the oral and pharyngeal jaws[14,25–27]. There is also a need in the field to assess integration at multiple taxonomic levels, as patterns at higher, macroevolutionary levels (i.e., between genera) may or may not be related to those at lower, microevolutionary levels (i.e., within species) or developmental levels (i.e., within individuals), owing to distinct genetic and evolutionary phenomena influencing the manifestation of patterns at each level. For example, examining associations *across* cichlids may reveal unique combinations of oral jaw and pharyngeal jaw traits in a minority of taxa that, while their jaws appear more decoupled (i.e., residing far from a line of best-fit), may still be integrated but represent a reversal of the direction of the correlation (Fig. 1a; association follows the perpendicular trajectory, light gray arrows of light gray ellipse). Examining jaw associations *within* cichlid taxa can complement such higher-level analyses by comparing (1) the strength of integration between species (Fig. 1b; narrow distribution versus wide distribution around best-fit line), and (2) the aspects of the anatomy that are co-varying in each lineage. For the purposes of this project, when we describe the relationship between the LOJ and LPJ of cichlids as 'integrated', we are referring to the strength of correlation between jaw shapes; however, we also provide descriptions of anatomical (co)variation so that patterns of integration may be assessed and compared between evolutionary levels.

There is evidence to suggest that cichlids can experience some variation in the degree of integration between their oral and pharyngeal jaws, and that integration itself may be dependent on feeding ecology[14]. Specifically, despite the perceived ability of cichlids to free themselves from mechanical constraints, cichlids that hunt elusive prey typically pair slender, mobile oral jaws with gracile pharyngeal jaws, while cichlids that feed on algae or other tough foods typically pair robust, compact oral jaws with strong pharyngeal jaws. This form-function relationship suggests feeding ecology may be the predominant force that determines integration between jaws. An alternative explanation is that the oral and pharyngeal jaws of cichlids are integrated as a consequence of some intrinsic genetic or developmental constraints such as pleiotropy[28], and that this precludes many cichlid taxa from exploring all the available functional combinations of jaw morphologies. Asserting this scenario posits that a handful of taxa, typically with specialist diets, have broken free of these constraints or exhibit unique oral-pharyngeal jaw association trajectories (e.g., lepidovores, molluscivores, ovivores etc.), and suggests decoupled oral and pharyngeal jaws are an exception, and not the rule.

Here we characterize the integration between cichlid lower oral jaw (LOJ) and lower pharyngeal jaw (LPJ) shapes at the genetic, phenotypic, and evolutionary level. We test the hypothesis that the oral and pharyngeal jaws of cichlids represent distinct phenotypic and genetic modules that would permit the exploration of independent evolutionary trajectories directed by selection. We employ an array of phylogenetic comparative methods, statistical genetics, and molecular approaches to demonstrate how certain aspects of shape variation in the oral and pharyngeal jaws of cichlid fishes are evolutionarily and genetically coupled. Our data roundly support the idea that the oral and pharyngeal jaws are integrated. Furthermore, we uncover a set of linked candidate genes that may contribute to this integration. By taking a multifaceted approach that assesses several levels of organismal diversity, from genes through morphology, function and resource-use, we gain a more holistic understanding of how

adaptive variation is generated under constraint, and how constraints, or the lack-thereof, can contribute to our understanding of diversification[29,30].

## Results

**Macro- and micro-evolutionary integration between jaw complexes.** We examined phenotypic associations between the lower oral and pharyngeal jaws (LOJ and LPJ, respectively) of 88 cichlid species from across Africa, primarily sampling from lakes in the East African Rift Valley: lakes Malawi, Tanganyika, and Victoria (Supplementary Data 1). We characterized jaw shapes based on 107 individuals using 3D geometric morphometrics by placing landmarks in positions that capture functionally (e.g., bony processes, sutures, etc.) and developmentally (e.g., distinct cellular origins) important aspects of morphology, including placing mirrored landmarks across midlines to gain symmetric configurations (Fig. 1e, Supplementary Fig. 1). We conducted a Procrustes superimposition, removed the effects of allometry to account for size differences, and then removed the effects of asymmetry to account for developmental noise. We performed a two-block partial least squares (PLS) analysis on the species mean landmark configurations and corrected for phylogenetic non-independence using a Bayesian time-calibrated tree[31]. We found the LOJ and LPJ were evolutionarily correlated (r-PLS = 0.482, $P = 0.002$, effect size ($Z$) = 2.585), but some taxa, particularly those with unique diets and/or modes of feeding, appeared to deviate from the best-fit line, indicating lower levels (or different patterns) of integration between jaws (Fig. 2a). Indeed, we found numerous taxa, typically from Lake Malawi, whereby covariation between the LOJ and LPJ appeared much different relative to other African cichlids. Taxa placed far from the best-fit line either (1) exhibited a specialized feeding morphology to better exploit an foraging niche shared with many competitors (i.e., *Labeotropheus*, algae; *Copadichromis*, zooplankton; *Taeniolethrinops*, insects), or (2) exhibited a specialized feeding morphology to take advantage of a more challenging food source (i.e., *Trematocranus*, snails). However, not all taxa that consume specialized prey were far from the best-fit line; *Pungu*, (primarily a sponge-feeder) and *Perissodus* (a scale-feeder), while exhibiting specialized feeding apparatuses to consume such prey, exhibited a relationship between their LOJ and LPJ that was in-line with other African cichlids (Supplementary Fig. 2). We also noted, that while Malawi cichlids exhibit a range of LOJ-LPJ relationships (from weak to strong), most Tanganyikan cichlids reside close to the best-fit line. However, when we examine the strength of integration in the Tanganyika group ($n = 29$, r-PLS = 0.698, $P = 0.001$, $Z = 2.954$) and Malawi group ($n = 40$, r-PLS = 0.541, $P = 0.020$, $Z = 2.155$), despite Tanganyika cichlids exhibiting higher $Z$-scores, consistent with stronger integration, a statistical comparison between groups finds no significant difference ($Z$ pairwise = 1.188, $P = 0.235$). Comparisons between Tanganyikan and Malawi cichlids should not be influenced by sampling bias, as principal components analyses (PCA) on the LOJ and LPJ landmark data (Supplementary Data 2 and 3) showed that our sampling of Tanganyikan cichlids includes many species with extreme morphologies that reside at the outer edges of LOJ and LPJ morphospace (Supplementary Fig. 3). Indeed, cichlids from Lake Tanganyika exhibited similar LOJ morphological disparity (Malawi Procrustes variance (PV) = 0.074; Tanganyika PV = 0.057, $P = 0.253$) and greater LPJ morphological disparity (Malawi PV = 0.015; Tanganyika PV = 0.023, $P = 0.012$), relative to cichlids from Lake Malawi. Taken together, this indicates that while Tanganyikan cichlids exhibit comparable (i.e., LOJ), or greater (i.e., LPJ) morphological variation compared to Malawi cichlids, covariation between LOJ and LPJ shapes was generally similar between groups.

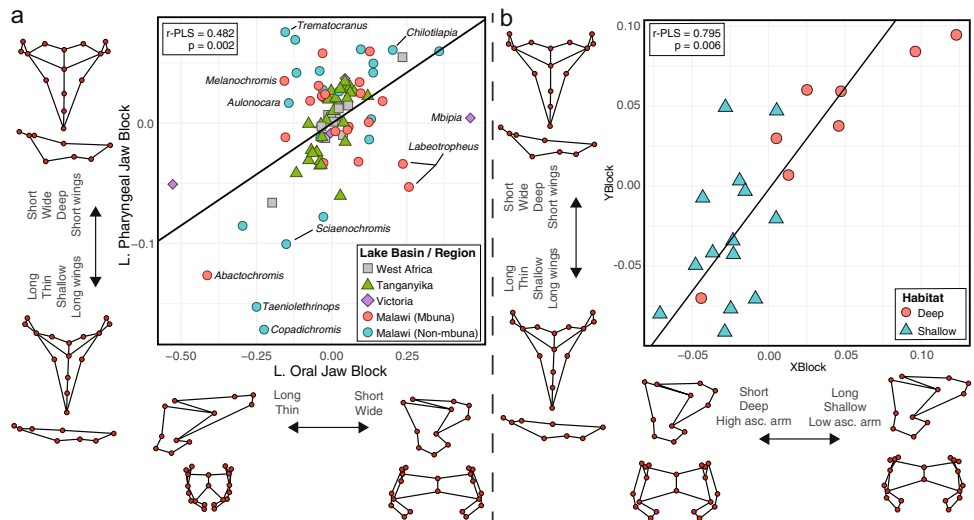

**Fig. 2 Phylogenetic two-block partial least squares analysis to assess macroevolutionary associations between lower oral and pharyngeal jaws. a** Jaw shape associations across a broad sample of African cichlids ($n = 88$). Taxa from Lake Malawi are placed into two groups based on phylogenetic position: an mbuna 'rock-dwellers' group, and a non-mbuna group consisting of the utaka 'sand-dwellers' alongside other benthic species[88]. **b** Jaw shape associations across the *Tropheops sp.* species complex from across a depth gradient ($n = 22$). Oral and pharyngeal jaw wireframes denote morphologies at either end of the correlational axis. Source data are provided as a Source Data file.

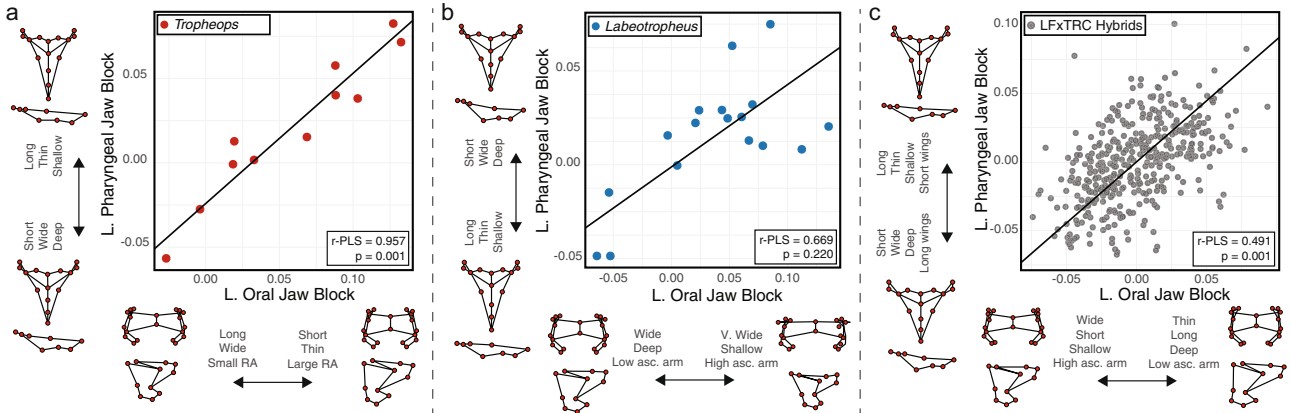

**Fig. 3 Two-block partial least squares analysis to assess microevolutionary associations between lower oral and pharyngeal jaws. a** Shape associations among *Tropheops sp.* "red cheek" (TRC) individuals ($n = 11$). **b** Shape associations among *Labeotropheus fuelleborni* (LF) individuals ($n = 17$). **c** Shape associations among members of a hybrid cross between TRC and LF ($n = 409$). Oral and pharyngeal jaw wireframes denote morphologies at either end of the correlational axis. Source data are provided as a Source Data file.

We next investigated the degree of integration at lower taxonomic levels. First, we analyzed the jaws of cichlids within the *Tropheops* species complex from Lake Malawi that is diverse and known to partition habitat by depth[32,33]. While *Tropheops* exhibited strong integration between jaws in on our macroevolutionary assessment, species within this genus occupy a broader niche space. Investigating integration within such a species complex provided an opportunity to understand whether habitat differences could lead to differences in integration between jaw complexes. Using the same landmarking procedure as described above, we characterized shape variation in the LOJs and LPJs of 22 wild-caught *Tropheops* taxa from 60 individuals, concentrating on members from localities across the southern portion of Lake Malawi (Supplementary Data 4). We again performed a two-block PLS analysis on the mean landmark configurations and accounted for phylogenetic non-independence using an amplified fragment length polymorphism tree[33]. Again, we found the LOJ and LPJ were evolutionarily correlated (r-PLS = 0.795, $P = 0.006$,

$Z = 2.521$), indicating jaw integration does not appear to vary by habitat (Fig. 2b).

Finally, we measured and compared integration between a species pair that exhibited relatively strong versus weak covariation between LOJ and LPJ shapes in our macroevolutionary assessments, *Tropheops sp.* 'red cheek' (TRC, relatively stronger integration) and *Labeotropheus fuelleborni* (LF, relatively weaker integration). Using the same landmarking protocol we performed separate two-block PLS analyses between LOJs and LPJs of LF and TRC (Supplementary Data 5). Notably, we found strong and significant integration between jaw complexes in TRC ($n = 11$, r-PLS = 0.957, $P = 0.001$, $Z = 3.038$; Fig. 3a) relative to LF ($n = 17$, r-PLS = 0.669, $P = 0.22$, $Z = 0.794$; Fig. 3b). Further, we found the effect sizes of jaw integration within TRC and LF to be statistically distinct (Z pairwise = 1.678, $P = 0.047$). Altogether, our data support the assertion that the LOJ and LPJ are evolutionarily integrated at multiple taxonomic levels, but they also appear to indicate that certain taxa, such as *Labeotropheus*,

can more readily generate adaptive morphological variation in each jaw complex independently.

**Genetic basis for oral and pharyngeal jaw shape covariation**. To understand whether phenotypic covariation between the LOJ and LPJ is genetically determined we performed a quantitative trait loci (QTL) analysis to identify prospective genomic regions involved in jaw shape variation for both the LOJ and LPJ. Specifically, we extended an existing genetic cross between the more strongly integrated TRC and the more weakly integrated LF to the $F_5$ generation. Details of the pedigree may be found in[34] and in the supplement. For this experiment, we genotyped 636 $F_5$ hybrids and produced a genetic map containing 812 single-nucleotide polymorphisms (SNPs) spread across 24 linkage groups (Supplementary Data 6). With a total length of 1431 cM, our high-resolution linkage map contained a marker every 1.83 cM, on average, allowing us to leverage the increased number of recombination events that occurred to reach an $F_5$ population. We then characterized LOJ and LPJ shape in 409 $F_5$ hybrids using the same landmarking scheme described above, and performed a two-block PLS analysis. In concordance with our findings from natural populations, we documented an association between jaw complexes in this laboratory pedigree (r-PLS = 0.491, $P = 0.001$, effect size = 6.189; Fig. 3c).

We next performed a PCA on the hybrid landmark configurations to distill the data down to a series of orthogonal axes that best explain shape variation among individuals. We extracted the first two PCs from the LOJ and LPJ as each axis represented more than 10% of the shape variation (Supplementary Data 7; Supplementary Figs. 4 and 5). The first axis of the LOJ reflected more general variation in depth, width, and length of the element (41.8% of variation), while the second axis reflected more specific variation in the length of the ascending arm of the articular––the process for which jaw closing muscles attach (12.7% of variation). The first axis of the LPJ reflected width, length, and wing process size (33.7% of variation), while the second axis reflected depth and the size of the anterior keel – the process for which the pharyngeal jaw pharyngohyoideus muscle attaches and controls jaw adduction (14.2% of variation). We then utilized these PC scores as traits to run in our QTL analyses to investigate the genetic basis for variation in these structures.

**QTL mapping implicates pleiotropic control of LOJ and LPJ shape variation**. Integration between LOJ and LJP shapes in the $F_5$ predicts that this pattern of covariation will be reflected in the genotype-phenotype map. Specifically, we predict that we will find overlapping QTL for both jaws. We used a multiple QTL mapping (MQM) approach to test this prediction. Specifically, we performed QTL scans for all four traits and quantitatively assessed the evidence for significant QTL marker(s) using a permutation procedure that reshuffles the phenotypic data relative to genotypic data 1000 times to generate a null distribution, disassociating any possible relationship between genotype and phenotype, to then compare the empirical distribution against[35]. Once candidate QTL markers were identified, we calculated an approximate Bayesian credible interval to determine the region in which a potential candidate locus would reside. We uncovered a total of five QTL for LOJ traits, and four QTL for LPJ traits (Fig. 4a; Supplementary Data 8). While most QTL localize to different linkage groups, we also identified some QTL that colocalized. Two traits (LOJ PC1, LPJ PC1) share a marker on LG4, while three traits (LOJ PC1, LOJ PC2, LPJ PC1) colocalized to the same markers on LG7. These data are consistent with pleiotropy on LG7 and possibly LG4.

We then quantitatively assessed the evidence for pleiotropy using a likelihood ratio test (LLRT) to compare the null hypothesis of a common pleiotropic QTL to the alternative hypothesis that they are affected by separate QTL[36,37]. The overlap on LG4 at a single marker (43.57 cM) was deemed significant (LLRT = 1.85, $P = 0.03$), indicating that we can reject the null hypothesis and that these peaks likely represent separate QTL for each trait (Supplementary Fig. 6). The three traits that overlap on LG7 spanned two markers (19.12 cM–28.04 cM) and were all deemed non-significant (LOJpc1-LPJpc1: LLRT = 0.02, $P = 0.66$, Fig. 4b; LOJpc2-LPJpc2: LLRT = 0.20, $P = 0.41$, Fig. 4c), leading us to accept the null hypothesis and conclude that this interval likely contains a single pleiotropic QTL. Whether a single gene, or multiple closely linked genes drive this pleiotropic signal requires a fine-mapping approach.

Notably, this locus on LG7 has been implicated previously in underlying LOJ and LPJ shape in another Lake Malawi cichlid cross between LF and *Maylandia zebra*[38,39]. *Maylandia* species, like *Tropheops*, were generally more integrated in our macro-evolutionary analysis (Fig. 2a), and thus another cross between LF and a species with higher integration values point to the same locus. This suggests that the genetic mechanism of integration may be conserved.

**Fine mapping identifies two candidate genes critical for bone formation**. To gain insights into which gene(s) may be pleiotropically regulating LOJ and LPJ jaw shape variation on LG7 we constructed a fine map with greater marker density to investigate genotype-phenotype associations with greater resolution. To that end, we anchored QTL intervals to particular stretches of physical sequence of the *Maylandia zebra* genome[40]. We then identified additional RAD-seq SNPs across the linkage group of interest and genotyped them in the $F_5$. Based on this we created two fine maps: the first spanned the entirety of LG7 group with an average spacing of around one marker every 490 kb (Supplementary Data 9), the second matched the QTL marker range revealed by the Bayesian credible interval analysis with an average spacing of around one marker every 180 kb (Supplementary Data 10). We also calculated $F_{ST}$ from a panel of wild-caught LF ($n = 20$) and TRC ($n = 20$), and primarily focused on $F_{ST}$ values of 1.0 that would indicate complete segregation of a SNP between LF and TRC. At every marker on our LG7 fine maps, we calculated the difference in the values of our three colocalized traits between those hybrids homozygous for the LF allele and those homozygous for the TRC allele.

We identified a small region on LG7 that exhibited large differences in the average phenotypic effect of those hybrids with either LF or TRC alleles. In our full LG7 map we identified a ~2 mb region (46.7 mb–48.7 mb) that peaked in all three traits (Fig. 4d; Supplementary Data 11). Notably, the traces of all three traits across our LG7 fine maps track together in an almost identical fashion. In our fine map that centered on the Bayes credible interval, we found evidence for both large phenotypic effects among all traits, and the presence of several $F_{ST}$ markers approaching or equal to 1.0 (Fig. 4e; Supplementary Data 12). One marker combined an $F_{ST}$ score of 1.0, indicating complete segregation of that allele between LF and TRC, with high average phenotypic effects across all traits (Supplementary Fig. 7). This SNP fell within an intron of *dymeclin* (*dym*), a gene that is necessary for correct organization of Golgi apparatus and controls endochondral bone formation during early development. *Dym* is critical for chondrocyte development and previous research using the zebrafish model found an expression pattern that spanned the presumptive mandibular and ceratobranchial regions at larval stages[41]. Mutations in this gene lead to profound effects

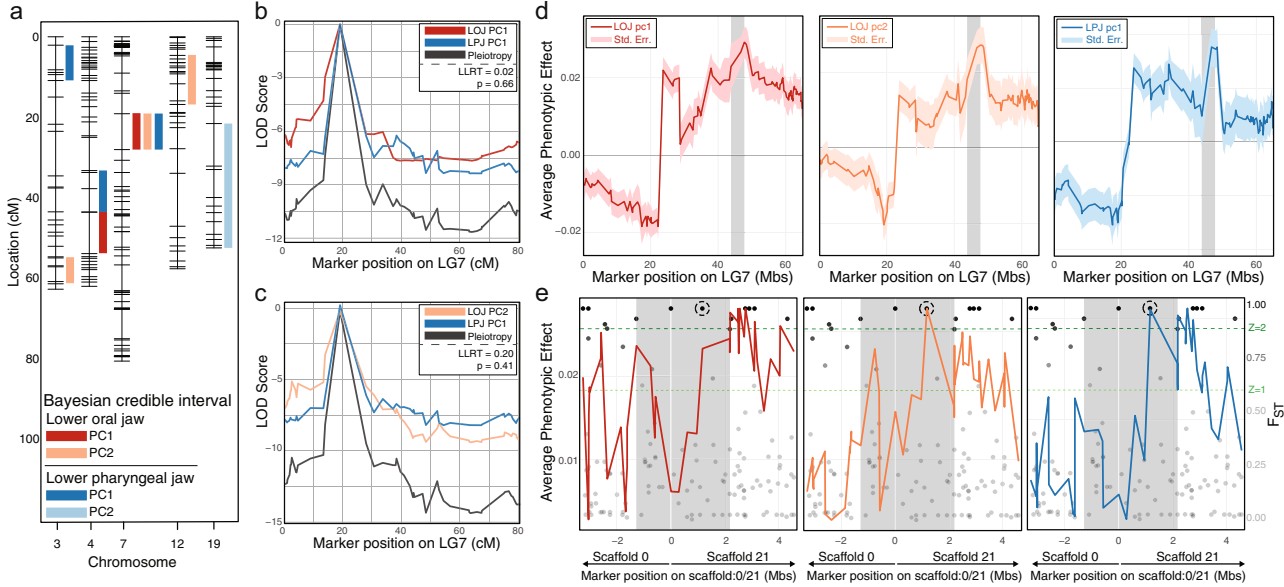

**Fig. 4 Genetic analyses to identify regions of the genome responsible for major changes in jaw shape.** All plots are based on 409 LFxTRC F₅ hybrids. **a** QTL analysis to identify positions in the genome most associated with each trait. **b** Pleiotropy analysis on linkage group seven to determine whether the oral jaw PC1 trait colocalizes to the same region as the pharyngeal jaw PC1 trait. Significance was determined using a likelihood ratio test (LLRT). **c** Pleiotropy analysis on linkage group seven to determine whether the oral jaw PC2 trait colocalizes to the same region as the pharyngeal jaw PC1 trait. Significance was determined using a LLRT. **d** Fine mapping all traits across the entirety of LG7. Values furthest from 0 reflect the largest differences between hybrids with LF and TRC genotypes at a given marker. We find peak genotype-phenotype association at ~50 mb that coincides with our Bayes credible interval (grey bar). Intervals that surround the average phenotypic effect line denote standard error of the mean. **e** Fine mapping all traits across the Bayes credible interval. Population level genetic diversity (F_{ST}) data are applied to the map (black dots) with the opacity of each SNP dependent on the degree of segregation between LF and TRC, with those falling above an empirical Z-score threshold of 0.6 determined to be significant, and those above 0.9 deemed highly significant (green lines). Within the credible interval there are four SNPs with F_{ST} values of 1.0, but a single SNP that falls within a genotype-phenotype peak residing within an intron of *dym* (black circle). Source data are provided as a Source Data file.

on the size and shape of bones due to misregulated chondrocyte development[42]. Just downstream (8 kb, Supplementary Fig. 7) of *dym* is *mothers against decapentaplegic homolog 7* (*smad7*), an antagonist of both TGF-β and BMP signaling and a suppressor of bone formation. As an inhibitory Smad, *smad7* negatively regulates genes from the BMP and TGF-β signaling pathways (i.e., *bmp-2, -4, -7, nodal*, etc.) that are known to shape phenotypic differences in the craniofacial skeleton across a wide range of taxa including cichlids[25,38,43], *Geospiza* finches[44,45], and *Anolis* lizards[46], primarily because these genes have the capacity to influence size in structures of trophic importance such as the mandible[47]. Both of these genes represent good candidates for controlling shape variation in the LOJ and the LPJ simultaneously. While two of the three traits peak at the *dym* SNP, when considering markers just outside the credible interval another peak is visible (especially for LOJ PC1) that sits close to *notch1a*, a gene involved in skeletal development and homeostasis. *Notch1a* is flanked by two fully segregated F_{ST} markers. The upstream marker is around ~60 kb from the promoter region, while the downstream marker resides around ~52 kb away from the gene within an intron of *kcnt1*, a gene involved in potassium channel development that appears to regulate brain function[48]. While *kcnt1* reflects a poor candidate gene for our analysis, the intronic SNP could act as a distant enhancer of *notch1a*. Thus, given the combined results from QTL and fine-mapping, *dym* and *smad7* represent strong candidates, but we cannot rule out *notch1a*.

**Correlated expression of key genes between LOJ and LPJ.** We used quantitative real-time PCR (qPCR) to assess and compare the expression levels of *dym*, *smad7*, and *notch1a* in the LOJ and LPJ of three *mbuna* genera from lake Malawi (*Tropheops* n = 6, *Labeotropheus* n = 8, *Maylandia* n = 8). We used *Labeotropheus*

and *Tropheops* to complement our quantitative genetic analysis, and all three taxa were represented in our phenotypic assessments of integration, permitting a comparison between macroevolutionary associations of the LOJ and LPJ with the underlying genetic architecture and expression for jaw complex correlation. We collected tissue samples from young juveniles of these four taxa, taking the LOJ and LPJ, alongside the caudal fin to act as an internal control, and performed a phenol/chloroform RNA extraction. We designed primers with high amplification efficiency (>90%) for our three genes (Supplementary Data 13), and used *β-actin* as our control gene. We calculated relative expression of the LOJ and LPJ using the 2^{-ΔΔCT} method[49], and compared expression across taxa and between tissues (Supplementary Data 14 and 15).

We initially compared tissue level expression levels between *Labeotropheus* and *Tropheops* and found small differences in *dym* expression, with LF typically exhibiting slightly higher levels (t-test LOJ t = 2.863, P = 0.014; LPJ t = 1.212, P = 0.249; Fig. 5a). These results are consistent with previous expression studies that demonstrated how *Labeotropheus* typically has up-regulated bone and collagen markers and as a consequence has greater bone deposition and a more robust craniofacial skeleton[50,51]. Expression level differences were also noted for *notch1a* and *smad7* (Fig. 5b-c); both showed reduced expression in LF, which is expected based on each genes role as negative regulators of bone formation[52,53]. While the differences between species were fairly small in *smad7* between taxa (t-test LOJ t = -1.869, P = 0.086; LPJ t = -0.359, P = 0.726), they were more notable in *notch1a* (t-test LOJ t = -1.947, P = 0.080; LPJ t = -3.221, P = 0.009). *Notch1a* is involved in skeletal remodeling, previous research has shown LF exhibits a minimal plastic response to environmental stimuli[51]. Thus, the relatively low expression of *notch1a* in

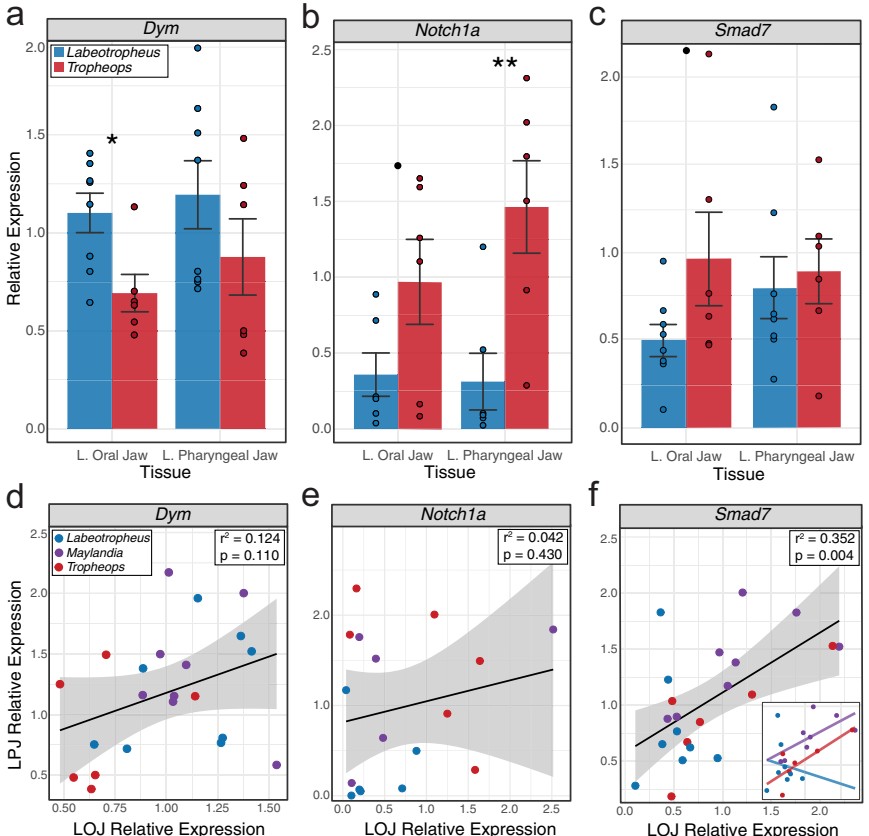

**Fig. 5 Comparing expression levels of *dym*, *smad7*, and *notch1a* via qPCR in the oral and pharyngeal jaws. a** *dym* bar plot; (**b**) *notch1a* bar plot; (**c**), *smad7* bar plot; (**d**), *dym* scatter plot; (**e**), *notch1a* scatter plot; (**f**), *smad7* scatter plot. **a–c** bar plots depict mean relative expression levels, error bars denote standard error. **d–f** Scatterplots depict relative expression levels of the LOJ and LPJ, error bounds surrounding the linear regression line denote standard error. **e** inset, linear regression for each genus. Three cichlid taxa were included: *Labeotropheus* $n = 8$, *Tropheops* $n = 6$, *Maylandia* $n = 8$. Bar plot significance determined via *t*-tests: ●$P < 0.10$, *$P < 0.05$, **$P < 0.01$. Source data are provided as a Source Data file.

*Labeotropheus* compared to *Tropheops* is consistent with this observation. While only representing a single life-history stage, the expression differences between species suggest that all three genes may underlie the development of species-specific shapes for the LOJ and/or LPJ. However, visualizing the data this way cannot speak to whether one or more of these loci underlie the covariation of the jaws.

To more explicitly address this question, we assessed the strength of correlation between LOJ and LPJ expression levels with all taxa. We found no evidence for correlated expression of *dym* between the LOJ and LPJ among taxa ($r^2 = 0.124$, $P = 0.11$; Fig. 5d), nor *notch1a* ($r^2 = 0.042$, $P = 0.43$; Fig. 5e), whereas we found a conspicuously strong correlation for *smad7* ($r^2 = 0.352$, $P = 0.0036$; Fig. 5f). These findings indicate that while *dym* and *notch1a* expression may be different between taxa and tissues, they do not appear to be strong candidates for driving coordinated change in the LOJ and LPJ. Alternatively, *smad7* represents a more robust candidate for regulating the covariation between these two structures.

Another pattern that emerged from these data is that not all taxa appear to be contributing equally to correlations in gene expression. For *smad7*, *Tropheops* individuals fall close to the trend line, while *Labeotropheus* individuals do not. This observation is notable, because it agrees with our macro- and microevolutionary analyses, which demonstrated that *Labeotropheus* exhibits weak integration between the LOJ and LPJ compared to *Tropheops* (Figs. 2a, 3a, b). When quantitatively compared, we observed evidence of more correlated *smad7*

expression between the jaws in *Tropheops* ($r^2 = 0.641$, $P = 0.056$) and *Maylandia* ($r^2 = 0.452$, $P = 0.068$), but not in *Labeotropheus* ($r^2 = 0.02$, $P = 0.74$). Alternatively, *dym* expression in the LOJ and LPJ was not correlated in any taxa (*Tropheops*, $r^2 = 0.076$, $P = 0.60$; *Maylandia*, $r^2 = 0.052$, $P = 0.59$; *Labeotropheops*, $r^2 = 0.188$, $P = 0.28$). These data raise the interesting possibility that LF has evolved a mechanism to independently modulate expression of *smad7* between the LOJ and LPJ, and that this could represent a means to facilitate the decoupling of jaw shape variation (Fig. 5f, inset). Operationally, this may involve varying expression in one jaw system, over the other. For example, *Smad7* expression in the LOJ of *Labeotropheus* appears more constrained relative to *Tropheops* (Bartlett's $K^2 = 4.74$, $P = 0.029$), while expression in the LPJ is similarly variable to *Tropheops* (Bartlett's $K^2 = 0.048$, $P = 0.826$). Taken together, our expression data implicate *smad7* as a putative pleiotropic regulator of LOJ and LPJ shape variation in cichlids, an observation that deserves future exploration. Further, *dym* and *notch1a* remain candidates for regulating variation in LOJ and LPJ shapes, especially given their broad roles in craniofacial bone development, regulation, and homeostasis.

## Discussion

In spite of major advances in connecting genotype to phenotype, a considerable challenge in the field of evolutionary biology remains linking changes in the genome, to patterns of phenotypic evolution. One such open-ended question involves the origins of

genetic, developmental, or functional constraints and their impact on evolutionary potential[54]. These constraints ultimately influence the availability of variation for natural selection to act upon, driving evolution in certain directions while limiting it in others (Fig. 1a). There are a growing number of empirical examples of constraints impacting evolvability, which have been described at a variety of levels, including developmental (e.g., the ubiquitous pentadactyl limb of tetrapods[55]), functional (e.g., force-velocity trade-offs in bird feeding[56]), and evolutionary (e.g., reduced disparity of marsupial forelimbs owing to a climbing neonate[57]). In an environment with ample ecological opportunity, breaking from these constraints can lead to a rapid increase in morphological diversity and the occupation of new, previously inaccessible, ecological niches. For example, the transition to powered flight in bats limited body size, and thus eye size, making the capture of insect prey more difficult; however, some bats circumvented this constraint by developing a specialized auditory and echolocation system, removing their sole reliance on vision for hunting[58]. Similarly, the release of genetic constraints within *Labeotropheus* cichlids permitted the development of a hypertrophied snout via an expansion of ligamentous and connective tissues to increase the efficiency of cropping filamentous algae from rocks[43,59].

Another consequence of genetic, developmental, and functional constraints is phenotypic integration, whereby seemingly independent traits vary in unison. While determining the origins of integration has proved difficult (i.e.,[60]), trait integration can facilitate or limit morphological evolution depending on the direction of selection[2]. While phenotypic integration is a population-level metric, integration can have far-reaching effects on macroevolutionary processes such as taxonomic diversification, extinction, and morphological evolution[3,7]. Indeed, there is an emerging consensus that phenotypic integration can facilitate rapid and coordinated trait evolution in teleosts[5,61,62], and may reflect a general trend across organisms[4,63,64]. Within cichlids, a lack of functional integration between oral and pharyngeal jaws has enabled the separation of prey capture from prey processing, a mechanism that is considered key to their success[15]. However, we provide evidence that African cichlid trophic evolution has generally occurred within the context of a coupled jaw system, a trend that was also recently noted in New World cichlids[14]. We demonstrate further that LOJ-LPJ integration may be due, at least in part, to pleiotropy, a phenomenon that is commonly theorized to determine patterns of covariation and limit evolvability (i.e.,[9,65] but see[60]). We postulate that integration between LOJ-LPJ shapes in cichlids is not a constraint, but rather an attribute that promotes rapid shifts in foraging niche[33,66], an ability that should be particularly advantageous in young and/or dynamic environments similar to the East African Rift-Valley. If true, then LOJ-LPJ integration may be actively selected for, perhaps via pleiotropy. We also note the variation in the strength of LOJ-LPJ integration across rift lake cichlids, and that some taxa particularly foraging specialists (i.e., *Labeotropheus*, *Trematocranus*), exhibited weaker integration. This observation is consistent with the idea that pleiotropy itself can evolve, and the strength of integration that pleiotropy may create between traits is unlikely to be uniform across clades (reviewed in[28]). Thus, integration between the jaws of cichlids should not be considered a binary choice between "coupled" or "decoupled," but rather exist along a skewed distribution[14], with most taxa exhibiting tight integration between the jaws, and a subset of foraging specialists exhibiting less (or differently) integrated jaws. These observations are consistent with the idea that foraging niche partitioning may underlie the evolution LOJ-LPJ integration. We predict that had we used a cross between two species that both exhibited strong jaw integration (i.e., *Tropheops* and *Maylandia*), more pleiotropic loci would have been uncovered. Understanding the mechanisms

behind how *Labeotropheus* and other specialist cichlid taxa have evolutionarily decoupled their oral and pharyngeal jaws presents an interesting avenue for future exploration. More generally, documenting the genetic mechanisms behind jaw (de)coupling will be essential to understanding how the immensely diverse trophic morphology of cichlids originated and became a critical catalyst of the African cichlid radiations[67].

## Methods

**Specimen Collection.** Specimens used in our analyses were either imported directly from the wild, obtained through the aquarium trade and raised in the authors' fish facility, loaned from museums, or downloaded from online collection repositories. For our macroevolutionary analysis, we used 107 individual cichlids representing 88 taxa. We scanned 42 individuals representing 23 taxa, with trait values from most taxa based on measures taken from three individuals. We also downloaded 65 individuals representing 65 taxa from the online repository morphosource.org (Supplementary Data 1). For our microevolutionary study, we used 58 wild-caught *Tropheops* species representing 23 taxa, with 22 taxa present in the tree (Supplementary Data 4). When comparing jaw integration between LF and TRC, individuals were obtained from wild-caught and lab-raised populations. We scanned a total of 28 individuals: 17 LF and 11 TRC (Supplementary Data 5). Our hybrid mapping population between LF and TRC was generated in the lab and comprised 409 individuals (Supplementary Data 6 and 7). All experiments involving animals were performed in compliance with the Institutional Animal Care and Use Committee at UMass Amherst (#2018-0094 to RCA).

**uCT Scanning.** We performed all µCT scanning using an X-Tek HMXST 225 (Nikon Corporation). All scans were acquired at 25-35 micron resolution using 80–125 kV and 75–120 µA. We extracted z-stack images and segmented the hard tissues using Mimics (v19 Materialise NV), before exporting the 3D models to Geomagic 2014 (v1.0 3D Systems). We then used Geomagic to digitally dissect the lower oral and pharyngeal jaws from the whole organism ready for 3D geometric morphometrics.

**Geometric morphometric data collection.** We used a series of fixed landmarks (LMs) to characterize shape information by placing LMs on both the left and right sides in both jaws, which allowed us to separate the symmetric from asymmetric components of shape variation (Supplementary Fig. 1a, b; Supplementary Data 16). We placed a suite of LMs at key functional and developmental positions (i.e., processes, muscle insertion points, sutures etc.) on the lower oral jaw and the lower pharyngeal jaw (Supplementary Fig. 1c, d). We used a total of 20 fixed LMs on the lower oral jaw and 15 fixed LMs on the lower pharyngeal jaw and landmarked both jaws on the left and right sides across the midlines to permit later corrections for asymmetry, a product of developmental noise. All jaws sampled in this study were digitized based on the same landmark configuration. A single individual (AJC) performed all landmark digitizing using Landmark Editor (v3.0)[68] to eliminate inter-observer error. All subsequent morphometric and genetic analyses were performed using *R* (v4.0.1) unless otherwise stated[69].

For each of our four major analyses (macroevolutionary, microevolutionary, parentals, hybrids) we performed a Procrustes superimposition to remove the effects of translation, scaling, and rotation in our oral and pharyngeal jaw landmark configurations using the gpagen function in *geomorph* (v4.0.0)[70–72]. Following Procrustes superimposition, if multiple individuals were digitized from a single species in the macroevolutionary (across African cichlids) and microevolutionary (across *Tropheops* sp.) components of this study, we calculated a species mean landmark configuration. To assess the relationship between jaw size and shape in our landmark configurations we performed a Procrustes ANOVA between the centroid size and shape of each individual jaw using the *geomorph* procD.lm function or the procD.pgls function when comparing among species. We observed a small but significant degree of jaw allometry in many collections (Supplementary Data 17), and therefore extracted residual landmark coordinates from the Procrustes ANOVA to provide allometry-free jaw configurations for use in subsequent analyses.

To assess and correct for fluctuating asymmetry in our allometrically corrected lower oral and pharyngeal jaw landmark configurations we used the bilat.symmetry function in *geomorph*. All structures were found to exhibit significant degrees of directional asymmetry (Supplementary Data 18). We then extracted landmark configurations from both jaws based on the symmetric components of variation for use in each future analysis. Isolating the symmetric component of landmark variation should reduce the amount of developmental noise in our data and increase our ability to detect subtle genetic signals.

To examine the range of LOJ and LPJ morphologies among our African cichlids we performed a Principal Components Analysis (PCA) that reduced each jaw landmark dataset into a series of orthogonal axes that best describe variation among cichlid taxa (Supplementary Fig. 3; Supplementary Data 2 and 3). We performed PCA using the gm.prcomp function in *geomorph*. Given the large volume of morphospace occupied by those cichlids from Lakes Malawi and Tangyanika, we directly compared cichlid disparity between these lakes using

Procrustes variance. We performed a phylogenetically corrected disparity analysis using the `morphol.disparity` function in *geomorph*.

**Lower oral and pharyngeal jaw shape correlations**. We examined the degree of association between our allometrically- and symmetrically corrected lower oral and pharyngeal jaw shapes using partial least squares (PLS) analysis. When assessing the degree of jaw shape covariation among taxa from across the rift lakes we corrected for the phylogenetic non-independence of our jaw traits by using a Bayesian time calibrated tree[31], and incorporated that into our PLS using the `phylo.integration` function in *geomorph* to examine the degree of jaw association under a Brownian model of evolution[71,73]. Similarly, we used the same routines to conduct a phylogenetically corrected PLS on our collection of Lake Malawi *Tropheops* species from across deep and shallow habitats using an amplified fragment length polymorphism tree[33]. To assess the degree of jaw covariation within our two taxa of interest, TRC, and LF, and also within F5 hybrids of those species we again used PLS via the `two.b.pls` function in *geomorph*[74,75]. To compare integration between cichlids from Tanganyika and Malawi, and between LF and TRC, we used the `compare.pls` function in *geomorph* to statistically compare the effect sizes of our PLS analyses. While we list the correlation coefficient (r-PLS) and the multivariate effect size in all cases, there is no direct correspondence between the dispersion of projected scores and variance explained by axes, as in a PCA, meaning there is not a direct analog of total variation explained to total covariation explained in the PLS[76].

**Quantitative Trait Loci**. To begin constructing our genetic map we first crossed a wild-caught LF female from Makanjila Point with a TRC male from Chizumulu Island. We then interbred the resultant full-sibling F1 family to produce an F2 population for genotyping and the production of a genetic map[34]. Next, we performed further intercrosses by randomly interbreeding individuals from different families up to the F5 generation (*n* = 636). The additional intercrossing allowed this F5 hybrid population to undergo more recombination events, thus increasing the resolution of mapping intervals. We extracted and sequenced genomic DNA from caudal fin clips of 636 F5 hybrids following standard RAD-seq procedures[34,77]. As the F5 hybrid population did not exhibit a tractable pattern of Mendelian inheritance, we used a genetic map derived from the F2 generation of the same pedigree. In particular, we genotyped a subset of 812 evenly spaced genetic markers in the F5 population, and used this map for all future QTL analyses. Full details of the cross and construction of the map can be found in the supplement.

To extract shape information from our F5 hybrid oral and pharyngeal jaws we performed a principal component (PC) analysis that serves to reduce our landmark coordinate data to a series of orthogonal axes that best describe shape variation among individuals. The first axis, or component, reflects the greatest degree of shape variation change in the jaws, and subsequent axes describe progressively less shape information. We used the `gm.prcomp` function in *geomorph* to perform our PC analysis[71]. We extracted the first two PCs from each jaw to use as traits in our QTL analysis to identify regions of the genome that may be responsible for controlling variation in that trait. Subsequent PC axes explained <10% of the jaw shape variation. While PCs will typically capture large-scale changes in shape rather than, for example, the length of a particular process, we consider this a benefit as we are searching for loci that could bring about major correlated changes across jaw complexes.

To examine regions of the genome responsible for variation in lower oral and pharyngeal jaw shape we searched for putative loci using the multiple QTL mapping (MQM) approach[78] from *r/qtl* (v1.46-2)[79]. We performed a QTL scan via the `scanone` function to examine initial QTL peaks indicative of genotype-phenotype associations. We then manually added cofactors to the model based on the location of QTL peaks before performing a MQM analysis using the `mqmscan` function. As more cofactors were added, the `mqmscan` function determined cofactor fit via maximum-likelihood backward elimination. Cofactors were continually added to the model until we maximized the logarithm of odds (LOD) score. We then quantitatively assessed the significance of a QTL peak using the `mqmpermutation` function. The `mqmpermutation` function generates a null distribution of LOD scores at each marker by reshuffling the phenotypic data relative to genotypic data 1000 times[35]. QTL marker LOD scores that exceed a 5% LOD threshold are deemed significant. We then used the `bayesint` function to calculate an approximate Bayesian credible interval around a significant QTL marker for which a potential candidate gene responsible for shape variation would reside within.

Bayesian credible intervals that overlap among traits may be indicative of pleiotropic control. To quantitatively determine whether a pair of traits with overlapping credible intervals are pleiotropic or are associated with separate QTL, we used a likelihood ratio test[37]. This test compares the QTL position that maximizes the likelihood under pleiotropy at a common QTL location, to a pair of QTL positions that maximize likelihood under a model where QTL locations are distinct. The logarithm of the ratio between these two likelihood values is the test statistic. In this test, traits localizing to a common QTL reflect the null hypothesis, while traits localizing to distinct QTL reflect the alternative hypothesis. Significance testing of pleiotropy vs. separate QTL was obtained via parametric bootstrapping[80]. We performed the pleiotropy and bootstrapping tests using the `scan_pvl` function in the *qtl2pleio* (v1.4.3) R package[36,81].

**Fine Mapping**. Once we identified a candidate pleiotropic marker on linkage group seven we then identified additional RAD-seq SNPs in the F5 population and used them to map either the entirety of a linkage group (LG) or to a region defined by the Bayesian credible interval on a LG. We used the *Maylandia zebra* (MZ) genome to anchor QTL intervals to particular stretches of physical sequence along LG7[40]. We utilized two different MZ genomes in our fine-mapping as the most recent version (MZ UMD2a) enabled high-density sampling across LG7 (1 marker every 490 kb), while an older version (MZ UMD1.1) allowed us to combine mapping the credible interval with calculations of genetic divergence (1 marker every 180 kb). When mapping the credible interval the QTL marker range spanned two contiguous stretches of the MZ genome located on subsections of Scaffold 0 and Scaffold 21. We mapped larger regions of Scaffold 0 and 21 that contained the interval and extended beyond it on either side by ~200 kb. We used the VCFtools package (v0.1.16) to extract markers from a given linkage group or scaffold and recode into smaller VCF files[82], and then we used the *vcfR* (v1.10.0) package in R to translate these VCF files into genotypic information that is readable by *r/qtl*[83]. Once our fine maps were complete we merged in our trait data and examined the difference in average trait values between F5 hybrids with the LF allele and TRC allele at every marker position using the `effectsplot` function in *r/qtl*.

Finally, we assessed the degree of genetic divergence between wild-caught LF (*n* = 20) and TRC (*n* = 20) populations to identify regions in the genome that are nearing the alternate fixation of distinct alleles[34]. This entailed calculating F statistic estimates ($F_{ST}$) at each loci[84]. A large $F_{ST}$ value (~1.0) indicates high differentiation among populations at a given locus. We calculated Z-transformed $F_{ST}$ values ($zF_{ST}$) from markers across the genome to obtain thresholds for cichlid divergence. We plotted two thresholds on our fine maps ($Z = 2$; $Z = 1$) to illustrate markers undergoing divergence (Fig. 4e). In our cichlid populations, a Z-score of 1 equated to an $F_{ST}$ score of ~0.6, a value similar to that observed across other cichlid populations[85].

**Quantitative real-time PCR**. We performed qPCR to assess expression level differences in three genes of interest and to examine the correlation in expression levels between lower oral and pharyngeal jaws. We focused on three well-studied genera in the lab, *Labeotropheus*, *Tropheops*, and *Maylandia*, that exhibit a continuum of oral and pharyngeal jaw shapes. Expression level analyses were performed on *Labeotropheus fuelleborni* (LF, *n* = 8), *Maylandia callainos* (MC, *n* = 8), and *Tropheops kumara* (TK, *n* = 6) obtained from the aquarium trade. All animals were juveniles when sacrificed (LF standard length (SL), 40.4 mm–48.2 mm; MC SL, 44.9 mm–54.6 mm; TK SL, 44.3 mm, 61.2 mm (Supplementary Data 19). See the supplement for more information on fish facility conditions and animal husbandry protocols.

We dissected out the lower oral and pharyngeal jaws of our cichlids and took a clip of the caudal fin to perform phenol/chloroform-based RNA extractions. We performed a reverse transcriptase PCR on our extracted RNA to produce complementary DNA (cDNA) for qPCR. We designed primer pairs for *dym*, *notch1a*, and *smad7*, and used a previously published primer sequence for β-actin as our control gene[43]. We then checked primer efficiency using the standard curve method. We performed a qPCR on all our samples and calculated relative jaw expression following the $2^{-\Delta\Delta CT}$ method[49] using the *pcr* (v1.2.2) package in R[86]. We used *t*-tests to determine significance in expression level differences between LF and TRC, and then performed linear regression to examine the degree of correlation in expression levels. See the supplement for full details of the molecular analysis.

**Reporting summary**. Further information on research design is available in the Nature Research Reporting Summary linked to this article.

## Data availability
Additional data can be found in the supplementary information. RADseq data can be downloaded from Figshare, https://doi.org/10.6084/m9.figshare.15831834.v1. Source data are provided with this paper.

## Code availability
Scripts and raw data can be found on GitHub[87], https://github.com/andrewjohnconith/cichlid_OJPJ.

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

## Acknowledgements

We thank Greg Lin at Harvard for μCT access, Daniel Pulaski for assistance with segmentation software, and Prakrit Subba for contributing initial pilot data to inform the experimental design. This work was funded by the National Institutes of Health #R01DE026446 to RCA.

## Author contributions

A.J.C. and R.C.A. formulated research plan and experimental design. A.J.C. performed all statistical and genetic analyses. A.J.C. and R.C.A. wrote the manuscript.

## Competing interests

The authors declare no competing interests.
