## [Peer Review File · Nature Communications]

The cichlid oral and pharyngeal jaws are evolutionarily and genetically coupledREVIEWER COMMENTS

Reviewer #2 (Remarks to the Author):

This very interesting manuscript explores the degree of integration between oral and pharyngeal jaw systems in African Rift Lake cichlids, presenting morphological data broadly across species, across very closely related species, and in an F5 generation from a two species cross. These results are used to argue for significant integration between the jaw systems – across species, shape of the oral and pharyngeal jaws are significantly correlated. The study goes on to develop evidence for a role for pleiotropic effects of the *smad7* gene.

This is a terrific paper. The insights are significant. The work addresses a long-held notion that the presence of a second set of jaws in teleost fish (the pharyngeal jaw) decouples prey capture and processing functions and therefore allows some independent evolution of the two systems, possibly enhancing diversity as the potential exists for combinations of mechanical properties that would not be possible with a single jaw system doing both functions. This effect has previously been posited to be a major factor powering the diversification of cichlids. The present paper shows that in fact oral and pharyngeal jaws of cichlids are significantly integrated (although to be honest this result is not new, having been shown in a recent paper by Burress et al *Evolution* 74:950). A couple of examples were identified of lineages that have departed from that general trend and produced unexpected combinations of oral and pharyngeal jaw shape, which is interesting. Further, the authors produce considerable evidence for pleiotropic effects of at least one structural gene influencing both jaw systems, thus providing a developmental genetic mechanism for the integration.

Having been clear how much I like this ms, there are a number of minor issues that need attention. One or two of these might rise to the level of a major issue so I'll cover them first. The term 'integration' is used a lot in this manuscript. The problem is that this one word is used to describe several different phenomena. The result is a loss of clarity and sometimes internally inconsistent statements. As the authors point out on Line 367 phenotypic integration is a population-level metric, meaning it is a term that is applied alternatively to a group of individuals or a groups of species. The use of the term 'integration' seems awkward at several places in the ms (e.g. L95-103). Integration is measured as a correlation. If pharyngeal jaw morphology is highly correlated with oral jaw morphology then there is high integration. If a species shows a pairing of oral and pharyngeal jaw morphology that sits outside the overall correlation in the group, do we say it is decoupled or showing low integration? The wording is confusing (this is like saying that an outlier point in a plot of two correlated variables is 'uncorrelated') not least because the novel combination may in fact be caused by a developmental genetic switch to single gene that influences both system but now in a negative way rather than a positive way. The development of jaws could still be coupled but differently. Since the distinction of these levels is so central to this paper I beg the authors to be more clear with their meaning.

A second fairly substantial issue has to do with the partial regressions used to measure integration between oral and pharyngeal jaw shape. L131-133. In this analysis I am unclear on exactly what the r-square values represent. Is this the OVERALL correlation between oral jaw and pharyngeal jaw shapes? As I understand the method it is not. Or is this the strength of the correlation only between the axes within each jaw system that maximize covariation between jaw systems? (this is my understanding of this function in geomorph) If it's the former, $r^2 = 0.38$ seems quite strong. If it's the latter, it is entirely unclear how much variation in oral jaw shape is explained by pharyngeal jaw shape, and thus how decoupled their evolution is. Can you calculate the percent of TOTAL shape variation explained by this integration? I have not used this geomorph function so I confess I may be misunderstanding it. Regardless, this issue must be clarified in the paper.

Here are specific comments as I went through the ms.

L24-25. This is stated as though pharyngeal jaws are a Neoteleost synapomorphy? Pharyngeal jaws go back much further than that. Lauder has a paper on pharyngeal jaw function in *Polypterus*, for example. I've worked on gar and *Amia* PJs. Moray eels? There are studies out there on ostariophysan PJs.

L35. It's not clear what your answer is to the big prediction that decoupled jaws promotes diversification. It's like you answer a different question after posing the big prediction.

L74. Cichlids are certainly frequently associated with the concept of adaptive radiation, but the recent literature on them as an adaptive radiation emphasizes that the Cichlidae is actually made up of several adaptive radiations in the Rift Lakes and some smaller lakes but are not as rapidly diversifying in other habitats. I'm thinking of Ole Seehausen's work for example.

L72. Again, are cichlids generally characterized by rapid speciation as claimed here? Certainly this is true in the lake-specific radiations, but outside of lakes? Where is the evidence for rapid speciation outside of lake radiations? I am not sure this is an accurate generalization. Also, the wording here is off. The sentence is written as though the fish speciate rapidly within an array of niches. I think you mean speciation has resulted in many different niches or something like that?

L77. The modified pharyngeal jaw facilitates exploitation of more 'specialized' diets? I don't understand what you mean here. What is a 'specialized diet'? A narrow diet that includes few taxa? I just don't follow how the jaw novelty would have facilitated that. I also do not think that cichlids have been shown to have 'specialized' diets any more than other groups so I'm pretty confused by this.

L77-83. Also, I see how a pharyngeal jaw decouples prey capture and processing – but that seems to make the key innovation one for ray-finned fishes. What cichlids seem to have is a modified pharyngeal jaw. But it's not like that further decouples capture and processing. So how is the cichlid pharyngeal jaw a key innovation? Sorry, I should know this, but it's pretty confusing here.

L102. molluscivores, not moluscivores.

L400-411. The study includes a mixture of wild-caught and lab-reared individuals. It seems likely that feeding experiences during ontogeny help shape the adult phenotypes of these two jaw systems. It also seems that in the multi-species evaluations (e.g. Fig 2) some of the specimens aren't just lab-reared but are XX-generation individuals from highly inbred lab populations of these species. And two of the Victoria species were not even collected in Victoria. It's all quite worisome and not clear in the paper which fish were wild-caught and which are from inbred lab strains. And which species are the 'ecological specialists' and 'generalists' as referred to in line 121?

L400. Also, while the paper claims to be about cichlids, it's actually only about some Rift Lake cichlids, and chiefly Lake Malawi. I know there is a tendency to think of the Rift Lake cichlids as the big story but maybe the claims to describe what's going on in 'cichlids' should be tempered?

L124. "We characterized jaw shape...by placing landmarks on all functionally and developmentally important regions." This is a strong claim but does not seem to be backed up. How could you be sure that you have place landmarks on ALL FUNCTIONALLY AND DEVELOPMENTALLY IMPORTANT REGIONS? Should this be reworded?

L128. Here the analysis described is a two-block partial least squares. This is not how it is described in the methods section. Also, the methods are described here, repeating what is in the methods section. The redundancy should be cut down.

L144. To describe the most extreme outliers in the plot as having 'vastly more decoupled' jaw shape evolution seems to confuse different phenomena. It does suggest that they have not followed the same pattern of integration that most of your species have followed. But it may well be that, in a different way, shape evolution in that outlier lineage has been highly integrated (evolution of oral and pharyngeal shape was correlated). A simple example might be a reversal of the direction of the correlation –where pharyngeal jaws become longer and thinner while oral jaws become shorter and squatter. This might even be regulated by a single gene. The evolutionary integration of oral and pharyngeal shape might be just strong as in the rest of the group, but different in its directions through shape space.

L160-161. This is another spot where the 'integration' terminology issue leads to (my) confusion. This paragraph refers to 'integration' between jaw systems in two different ways. Indeed, is the same property being measured in each case? In the first it is the degree to which a species is an outlier in the broader group of species but in the second it is the correlation of shapes across individuals within species. They can't both be 'integration' can they?

L168. "Certain ecological specialists". I do not know what you mean that LF is an 'ecological specialist'. I sense that you feel that the outlier species in fig 2A exhibit a derived developmental pattern that allows them to achieve oral and pharyngeal shape combinations that are uncommon. Maybe these are associated with novel ecologies – you do not actually demonstrate that. But it would be another thing to show that it is associated with becoming an 'ecological specialist'.

L344. Just a suggestion here to only use the word 'major' once in this first sentence. The double use makes for awkward reading, especially in the first line of the discussion.

L372. Well, to be fair, this is the same result found by Ed Burress' 2020 paper on cichlid integration (Evolution 74:950), so the current study is not the first to show that oral and pharyngeal jaws of cichlids are, in fact, significantly integrated. Should that paper be cited here? Burress also found this phenomenon of a few outlier species being associated with novel trophic niches. I was the senior author on that paper so I apologize for blowing my own horn, but trying to be as objective as I can I genuinely feel that you are not giving enough credit to that Burress paper which established many of the major concepts presented in this paper and found much the same patterns of integration in another clade of cichlids.

L384-388. Again, a very similar result was found in Burress et al. 2020. It would be appropriate to cite that paper here.

-Peter Wainwright

Reviewer #3 (Remarks to the Author):

In this study, Conith and Albertson test for evolutionary coupling of the oral and pharyngeal jaws of cichlid fishes, mostly from Lake Malawi. It has previously been suggested that the functional decoupling

of these two jaw systems has contributed to the evolutionary "success" of cichlid and other neoteleost fish.

This study consists of two parts. In the first part, the authors report a PLS analysis of geometric morphometric data obtained for 40 species. This part is problematic for a number of reasons. First, using only 40 species (and for many of them a single specimen) does not do justice to the great taxonomic diversity of cichlids in the African Great Lakes. More importantly, these 40 species are in no way representative and do not even closely cover the phylogenetic, phenotypic or ecological diversity of cichlids in these lakes. Inferring "wide-spread integration between jaws" (see Abstract) from this limited data is not justified. Also, the much larger PLS-based analysis by Ronco et al. (published in Nature) seems to show different trajectories of the oral and pharyngeal jaws in cichlids from Lake Tanganyika.

In the second part, the authors present a massive and well-designed QTL experiment that led to the identification of a candidate gene exhibiting correlated expression between the two jaws. This latter part is novel and exciting. It is a pity, however, that the only "functional" evidence provided is a qPCR experiment. I thus do not think that it is justified to sell this study as "linking changes in the genome, to patterns of phenotypic evolution" (as the authors write in the beginning of the Discussion). In the end, what they provide is a list of QTL regions and candidate genes.

I did not get the part with the F_{ST} threshold of 0.57 (page 19). There is certainly no such thing as a common F_{ST} threshold for cichlids.

Reviewer #2 (Remarks to the Author):

This very interesting manuscript explores the degree of integration between oral and pharyngeal jaw systems in African Rift Lake cichlids, presenting morphological data broadly across species, across very closely related species, and in an F₅ generation from a two species cross. These results are used to argue for significant integration between the jaw systems – across species, shape of the oral and pharyngeal jaws are significantly correlated. The study goes on to develop evidence for a role for pleiotropic effects of the *smad7* gene.

This is a terrific paper. The insights are significant. The work addresses a long-held notion that the presence of a second set of jaws in teleost fish (the pharyngeal jaw) decouples prey capture and processing functions and therefore allows some independent evolution of the two systems, possibly enhancing diversity as the potential exists for combinations of mechanical properties that would not be possible with a single jaw system doing both functions. This effect has previously been posited to be a major factor powering the diversification of cichlids. The present paper shows that in fact oral and pharyngeal jaws of cichlids are significantly integrated (although to be honest this result is not new, having been shown in a recent paper by Burress et al *Evolution* 74:950). A couple of examples were identified of lineages that have departed from that general trend and produced unexpected combinations of oral and pharyngeal jaw shape, which is interesting. Further, the authors produce considerable evidence for pleiotropic effects of at least one structural gene influencing both jaw systems, thus providing a developmental genetic mechanism for the integration.

Response: We thank the reviewer for his careful and meticulous assessment of our manuscript. We detail below the changes we have made to better our analyses and writing based on these comments.

Having been clear how much I like this MS, there are a number of minor issues that need attention. One or two of these might rise to the level of a major issue so I'll cover them first. The term 'integration' is used a lot in this manuscript. The problem is that this one word is used to describe several different phenomena. The result is a loss of clarity and sometimes internally inconsistent statements. As the authors point out on Line 367 phenotypic integration is a population-level metric, meaning it is a term that is applied alternatively to a group of individuals or a groups of species. The use of the term 'integration' seems awkward at several places in the MS (e.g. L95-103). Integration is measured as a correlation. If pharyngeal jaw morphology is highly correlated with oral jaw morphology then there is high integration. If a species shows a pairing of oral and pharyngeal jaw morphology that sits outside the overall correlation in the group, do we say it is decoupled or showing low integration? The wording is confusing (this is like saying that an outlier point in a plot of two correlated variables is 'uncorrelated') not least because the novel combination may in fact be caused by a developmental genetic switch to single gene that influences both system but now in a negative way rather than a positive way. The development of jaws could still be coupled but differently. Since the distinction of these levels is so central to this paper I beg the authors to be more clear with their meaning.

Response: We agree that the terminology can become dense and the integration-decoupling discussions can become confusing. We have defined our meanings in the

introduction, and are careful with our language throughout, to help a reader understand how we are using these terms.

A second fairly substantial issue has to do with the partial regressions used to measure integration between oral and pharyngeal jaw shape. L131-133. In this analysis I am unclear on exactly what the r-square values represent. Is this the OVERALL correlation between oral jaw and pharyngeal jaw shapes? As I understand the method it is not. Or is this the strength of the correlation only between the axes within each jaw system that maximize covariation between jaw systems? (this is my understanding of this function in geomorph) If it's the former, $r^2 = 0.38$ seems quite strong. If it's the latter, it is entirely unclear how much variation in oral jaw shape is explained by pharyngeal jaw shape, and thus how decoupled their evolution is. Can you calculate the percent of TOTAL shape variation explained by this integration? I have not used this geomorph function so I confess I may be misunderstanding it. Regardless, this issue must be clarified in the paper.

Response: We directed this question to Prof. Dean Adams, one of the many architects of the geomorph package, and reproduce the e-mail chain between Drs Conith and Adams below (with permission of Prof. Adams). Including this conversation should hopefully benefit others with similar questions, given these reviewer responses are now published.

The key response comes directly from the Collyer and Adams 2021 paper and reads, "Additionally, although the singular values decay in a manner similar to eigenvalues, there is no direct correspondence between dispersion of projected scores and variance explained by axes, as with PCA." We have noted this in the text and directed a reader to this paper.

We also incorrectly squared the r-PLS values provided by the phylo.integration - two.b.pls functions, which likely led to some of the confusion surrounding this analysis as it made it appear like a 'coefficient of variation' value. We have corrected this back to the r-PLS value directly provided by the functions.

Additional important parts of the conversation with Prof. Adams went as follows:

"A. Conith: My understanding is that this [PLS] function is examining all dimensions of a multivariate shape trait to find those axes pertaining to maximum covariation. The reviewer asked me if there was a way to find out the total shape variation explained by the association.

Their reasoning was that an r value of 0.6 seems pretty strong if you are reporting the total variation, but it's a little more unclear how much variation in shape A is explained by shape B when looking at maximum variation, and thus what that means for how coupled or decoupled the evolution of each shape is.

Is there a way to calculate total variation using PLS (hopefully within geomorph)?

D. Adams: To answer your question, no, there is not a direct analog of total variation explained (say in PCA) to total covariation explained in PLS. Indeed, it seems desirable to explain things in these terms. However, what one is doing mathematically is describing covariance between the matrix product (projection) of X and Y. While that is described by singular values: and singular values are also used as a means to derive eigenvalues in PCA (and thus percent variation explained): the analogy stops there.

We discussed a bit in section 2.4 of our recent PACA paper in Methods in Ecology and Evolution [Collyer, ML, Adams, DC. Phylogenetically aligned component analysis. Methods

Ecol Evol. 2021; 12: 359– 372.]. There we were performing an ordination relative to phylogenetic signal: but mathematically, the analytics used are partial least squares. Thus, the discussion there regarding interpretation of axes and variation explained holds for your 2.b.pls. implementation.”

L24-25. This is stated as though pharyngeal jaws are a Neoteleost synapomorphy? Pharyngeal jaws go back much further than that. Lauder has a paper on pharyngeal jaw function in *Polypterus*, for example. I’ve worked on gar and *Amia* PJs. Moray eels? There are studies out there on ostariophysan PJs.

Response: We have changes this to ‘ray finned fishes.’

L35. It’s not clear what your answer is to the big prediction that decoupled jaws promotes diversification. It’s like you answer a different question after posing the big prediction.

Response: Thank you for pointing out the need for additional clarity. We have now changed this sentence to be more precise as to what our hypothesis is: that the jaws are decoupled. With this change we hope (and believe) the question now better matches the answer: that the jaws are not decoupled but integrated and thus the evolution of African cichlid foraging architecture seems to have occurred in the context of an integrated (not decoupled) jaw system.

L74. Cichlids are certainly frequently associated with the concept of adaptive radiation, but the recent literature on them as an adaptive radiation emphasizes that the Cichlidae is actually made up of several adaptive radiations in the Rift Lakes and some smaller lakes but are not as rapidly diversifying in other habitats. I’m thinking of Ole Seehausen’s work for example.

Response: We amended our writing to make it clear to a reader that the current diversity of cichlids is a product of multiple adaptive radiations, and not a single one. We have also included citations to some of Ole Seehausen’s work (see below) that dealt with this very question, and demonstrated how many of the cichlids within the ‘Great Lakes’ could speciate so rapidly.

Meier JJ, et al. Ancient hybridization fuels rapid cichlid fish adaptive radiations. *Nat Commun.* 2017. 8:14363.

Meier JJ, et al. The coincidence of ecological opportunity with hybridization explains rapid adaptive radiation in Lake Mweru cichlid fishes. *Nat Commun.* 2019. 10:5391.

L72. Again, are cichlids generally characterized by rapid speciation as claimed here? Certainly this is true in the lake-specific radiations, but outside of lakes? Where is the evidence for rapid speciation outside of lake radiations? I am not sure this is an accurate generalization. Also, the wording here is off. The sentence is written as though the fish speciate rapidly within an array of niches. I think you mean speciation has resulted in many different niches or something like that?

Response: As discussed above we have made it clear that rapid speciation is primarily a hallmark of lake radiations.

L77. The modified pharyngeal jaw facilitates exploitation of more ‘specialized’ diets? I don’t understand what you mean here. What is a ‘specialized diet’? A narrow diet that includes

few taxa? I just don't follow how the jaw novelty would have facilitated that. I also do not think that cichlids have been shown to have 'specialized' diets any more than other groups so I'm pretty confused by this.

Response: This was initially unclear. We now state that by foraging "specialist" we refer to either (1) species with unique jaws to better consume rare prey (e.g., snails, sponges) or (2) species with unique jaws to employ a novel mode of prey capture to better compete with taxa that consume the same prey item.

L77-83. Also, I see how a pharyngeal jaw decouples prey capture and processing – but that seems to make the key innovation one for ray-finned fishes. What cichlids seem to have is a modified pharyngeal jaw. But it's not like that further decouples capture and processing. So how is the cichlid pharyngeal jaw a key innovation? Sorry, I should know this, but it's pretty confusing here.

Response: We have now expanded upon our initial explanation to provide more anatomical information to distinguish the more 'typical' ray-finned fish pharyngeal jaws from those of cichlids. We also cite Hulsey (see below) to guide readers to an in-depth discussion of cichlid pharyngeal jaw evolution.

Hulsey, C. D. Function of a key morphological innovation: fusion of the cichlid pharyngeal jaw. *Proc. R. Soc. B Biol. Sci.* 273, 669–675 (2006).

L102. molluscivores, not moluscivores.

Response: Corrected.

L400-411. The study includes a mixture of wild-caught and lab-reared individuals. It seems likely that feeding experiences during ontogeny help shape the adult phenotypes of these two jaw systems. It also seems that in the multi-species evaluations (e.g. Fig 2) some of the specimens aren't just lab-reared but are XX-generation individuals from highly inbred lab populations of these species. And two of the Victoria species were not even collected in Victoria. It's all quite worisome and not clear in the paper which fish were wild-caught and which are from inbred lab strains. And which species are the 'ecological specialists' and 'generalists' as referred to in line 121?

Response: This is another good point. Specimens used in this study were either wild caught by an author (RCA), wild caught and housed in a museum collection, wild caught then lab reared (i.e., F₀), or lab raised (i.e., F₁-F₃). We have added a column on this in our supplementary tables (Table S1). It is worth noting that the most "inbred" fish were those used for pedigree mapping (i.e., F₅), and this was intentional to increase the number of recombination events and thus the resolution of fine mapping. These fish did not exhibit any deleterious signs of inbreeding, and therefore none of the other specimens are what we would consider highly inbred.

The Lake Victorian species were collected from surrounding lakes, but all surrounding lakes (i.e., Lake Albert, Lake Kyoga, Lake Nyasala etc.) are part of the Victorian basin and connect to each other (see Basiita et al. (2018) *PLoS ONE* 13(7): e0200001). We have now corrected our figure legends to better inform readers that cichlids are collected from lake basins (i.e., rivers and surrounding lakes that are connected), or outside of the Rift Valley (i.e., the west African designation).

We have now toned down the ecological specialist/generalist discussions, as we agree that they can become a point of contention.

L400. Also, while the paper claims to be about cichlids, it's actually only about some Rift Lake cichlids, and chiefly Lake Malawi. I know there is a tendency to think of the Rift Lake cichlids as the big story but maybe the claims to describe what's going on in 'cichlids' should be tempered?

Response: We have made it clear that much of our conclusions focus primarily on African cichlids.

L124. "We characterized jaw shape...by placing landmarks on all functionally and developmentally important regions." This is a strong claim but does not seem to be backed up. How could you be sure that you have place landmarks on ALL FUNCTIONALLY AND DEVELOPMENTALLY IMPORTANT REGIONS? Should this be reworded?

Response: We have re-framed this sentence such that the placement of landmarks can capture functionally and developmentally important structures, and provide examples of what we refer to.

L128. Here the analysis described is a two-block partial least squares. This is not how it is described in the methods section. Also, the methods are described here, repeating what is in the methods section. The redundancy should be cut down.

Response: We apologize but are unclear of the issue here. We state that we use a phylogenetically corrected PLS in both the results and the methods section when examining associations among taxa. We expanded upon the single sentence description from the results in the methods to describe the model of evolution used in the analysis alongside the specific package and function information. We believe this information is accurate, and the level of detail is appropriate in both places.

L144. To describe the most extreme outliers in the plot as having 'vastly more decoupled' jaw shape evolution seems to confuse different phenomena. It does suggest that they have not followed the same pattern of integration that most of your species have followed. But it may well be that, in a different way, shape evolution in that outlier lineage has been highly integrated (evolution of oral and pharyngeal shape was correlated). A simple example might be a reversal of the direction of the correlation –where pharyngeal jaws become longer and thinner while oral jaws become shorter and squatter. This might even be regulated by a single gene. The evolutionary integration of oral and pharyngeal shape might be just strong as in the rest of the group, but different in its directions through shape space.

Response: This is a really important point that deserves more discussion in the text. Certain cichlids may indeed simply follow a different (i.e., negative correlation) relationship that appears more unusual when placed in the same space as the vast majority of cichlids that exhibit a different type of jaw complex relationship (i.e., positive correlation). We now clarify this in the text, and also indicate that analyses at different taxonomic levels can help to clarify whether "outliers" are due to reduced levels or different patterns of integration.

For example, we highlight that when comparing integration between LF and TRC we found LF to be significantly less integrated (Figure 3a-b), which is consistent with our

macroevolutionary analysis where *Tropheops* specimens were close to the best-fit line, but *Labeotropheus* species were further away. We think the complementary nature of these analyses is important, illuminating, and not common in the field.

L160-161. This is another spot where the ‘integration’ terminology issue leads to (my) confusion. This paragraph refers to ‘integration’ between jaw systems in two different ways. Indeed, is the same property being measured in each case? In the first it is the degree to which a species is an outlier in the broader group of species but in the second it is the correlation of shapes across individuals within species. They can’t both be ‘integration’ can they?

Response: We may actually be detecting two distinct phenomena here. (1) *Labeotropheus* (alongside certain other cichlid taxa) exhibit a more unique combination of LOJ and LPJ traits and (2), the strength of integration within *Labeotropheus* is simply lower, and variation in the LOJ-LPJ association can be greater overall.

We have expanded upon this point in the introduction, tightened our wording to avoid confusion, and are more explicit about what we can (and cannot) infer from our data.

L168. “Certain ecological specialists”. I do not know what you mean that LF is an ‘ecological specialist’. I sense that you feel that the outlier species in fig 2A exhibit a derived developmental pattern that allows them to achieve oral and pharyngeal shape combinations that are uncommon. Maybe these are associated with novel ecologies – you do not actually demonstrate that. But it would be another thing to show that it is associated with becoming an ‘ecological specialist’.

Response: We are now more precise in our language and say “foraging specialists” and explicitly define this in the text. We also tone down discussions of specialists given this is a minor point in this paper and we do not explicitly test whether specialists reside closer or further from the best-fit line.

L344. Just a suggestion here to only use the word ‘major’ once in this first sentence. The double use makes for awkward reading, especially in the first line of the discussion.

Response: Corrected.

L372. Well, to be fair, this is the same result found by Ed Burress’ 2020 paper on cichlid integration (Evolution 74:950), so the current study is not the first to show that oral and pharyngeal jaws of cichlids are, in fact, significantly integrated. Should that paper be cited here? Burress also found this phenomenon of a few outlier species being associated with novel trophic niches. I was the senior author on that paper so I apologize for blowing my own horn, but trying to be as objective as I can I genuinely feel that you are not giving enough credit to that Burress paper which established many of the major concepts presented in this paper and found much the same patterns of integration in another clade of cichlids.

Response: Indeed! And in New World cichlids, which is a nice independent confirmation. We now cited Burress and more explicitly draw attention to other studies that have examined such questions.

L384-388. Again, a very similar result was found in Burress et al. 2020. It would be appropriate to cite that paper here.

Response: Citation added.

Thank you again for the thoughtful and thorough review!

Reviewer #3 (Remarks to the Author):

In this study, Conith and Albertson test for evolutionary coupling of the oral and pharyngeal jaws of cichlid fishes, mostly from Lake Malawi. It has previously been suggested that the functional decoupling of these two jaw systems has contributed to the evolutionary "success" of cichlid and other neoteleost fish.

Response: We thank the reviewer for their assessment of our manuscript and highlight a number of changes we made to our manuscript based on these comments.

This study consists of two parts. In the first part, the authors report a PLS analysis of geometric morphometric data obtained for 40 species. This part is problematic for a number of reasons. First, using only 40 species (and for many of them a single specimen) does not do justice to the great taxonomic diversity of cichlids in the African Great Lakes. More importantly, these 40 species are in no way representative and do not even closely cover the phylogenetic, phenotypic or ecological diversity of cichlids in these lakes. Inferring "wide-spread integration between jaws" (see Abstract) from this limited data is not justified. Also, the much larger PLS-based analysis by Ronco et al. (published in Nature) seems to show different trajectories of the oral and pharyngeal jaws in cichlids from Lake Tanganyika.

Response: To address these concerns, we have toned-down our language, and have more than doubled the sample size of the macroevolutionary component of this study, taking the number of taxa included in this analysis from 40 to 88. While we concede that many of our taxa are based on a single individual, the variation in cichlid jaw morphologies sampled from across Africa should be far greater than morphological variation within-species, as our sampling includes many extreme forms. Further, patterns of shape variation captured by our sampling (as determined by PCA and shown as Supplemental Information) are consistent with those published elsewhere. We feel that our sampling is amply sufficient to address our main question at the macroevolutionary level. Indeed, after doubling our sample size, the significance of our association between jaw complexes has only grown stronger.

We also stress that the macroevolutionary analysis was only one out of four analyses, each targeting a different taxonomic level. As discussed above, such an approach is rare in the field, and can be highly complementary and illuminating. Moreover, the results were highly consistent. Thus, by 'wide-spread' we are referring to the association between jaws persisting across levels – macroevolutionary, microevolutionary, and genetic. Altogether we analyzed the LOJ and LPJ of over 600 animals (~1200 bones total), and therefore feel confident in our results and conclusions.

Finally, the Ronco et al. comparison seems unjustified given the highly divergent goals of each project. Similarly, the statement that the Ronco paper "seems to show different trajectories of the oral and pharyngeal jaws [compared to our paper]" doesn't appear to be accurate. The way we read it the authors assesses associations between

phenotype and the environment, rates of morphological evolution, and genomic signatures of adaptive radiation – however, we perform none of these analyses. More specifically, Ronco and colleagues: 1, perform phenotype * environment associations, we perform phenotype * phenotype associations. 2, Assess the upper oral jaw in 2D (and it's association to an environmental variable), we assess the lower oral jaw in 3D (and it's association to a phenotypic variable - the lower pharyngeal jaw). 3, Examine the lower pharyngeal jaw in 3D (and it associations with environmental variables), we perform 3D LPJ shape analyses (and it's association with LOJ shape). 4, Perform GWAS-like analyses on Tanganyikan cichlids from across the lake, while our genetic analyses are based on a pedigree mapping population and examine specific genomic regions that may contribute to patterning correlated development of oral/pharyngeal jaw shape in African cichlids – in other words, between a mile-wide/inch deep vs. a mile-deep/inch-wide approach, we performed the latter, while Ronco et al. performed the former. Both studies are valuable, but our approaches and goals are vastly different, and we are therefore hesitant to make any comparisons between the two.

In the second part, the authors present a massive and well-designed QTL experiment that led to the identification of a candidate gene exhibiting correlated expression between the two jaws. This latter part is novel and exciting. It is a pity, however, that the only "functional" evidence provided is a qPCR experiment. I thus do not think that it is justified to sell this study as "linking changes in the genome, to patterns of phenotypic evolution" (as the authors write in the beginning of the Discussion). In the end, what they provide is a list of QTL regions and candidate genes.

Response: We apologize for the confusion, but feel that the reviewer has taken the above quote out of context. We in fact say that, "a considerable challenge in the field of evolutionary biology remains linking changes in the genome, to patterns of phenotypic evolution." By including the word "remains" we feel that we are pretty clear that our paper, while taking a step in the right direction, has not solved this challenge. This entire paragraph is setting out theory and open questions in the field that motivated our study. As we argue below, we do not feel that we are "selling" anything that our data don't support.

In terms of functional validation, we believe that the reviewer might be looking for some sort of genetic or small molecule manipulation. This is an approach we've taken in the past, and is reasonably straightforward when assessing patterns of *variation*, whereby a gene or pathway can be manipulated and the phenotypic effects can be compared to patterns of variation between species. This approach becomes far less tractable when seeking to validate genetic effects on the *covariation* between traits. Indeed, there really isn't a roadmap for this. We could take a reverse genetic approach and knockout *smad7* in zebrafish, for example, but this would not be terribly informative owing to the highly pleiotropic nature of this gene – e.g., of course multiple bones would be impacted, but so too would a multitude of other traits, and deciding which effects are primary versus secondary would be all but impossible. Functional studies using zebrafish would be further confounded due to the drastically different pharyngeal jaw apparatus (e.g., unfused) relative to those in cichlids.

It is not that we didn't think about ways to validate our mapping approach. We did. For example, with more time and resources, we could perform an ATAC-seq (or CHIP-seq) experiment to look for *cis*-regulatory changes in and around *smad7* in both LOJ and LPJ

tissues, and then search for homologous elements in zebrafish (or maybe a cichlid amenable to genetic manipulation – e.g., *A. burtoni*, *O. niloticus*), and then delete (or swap) putative enhancers via CRISPR/cas9 and query the effects on each jaw. We hope that the reviewer can appreciate that (while awesome!) such an endeavor would require much in the way of additional time and money, but *this is* the experiment that would be required to truly validate the mapping results.

We would argue further that a quantitative assessment of correlated gene expression is quite novel, and indeed more informative (as a means of validation) than the outcome of a relatively simple loss-of-function approach. When taken together with the mapping and phenotypic analyses, we have documented associations between jaws (1) from different genera across Africa, (2) from species within a genus, (3) from individuals within species, (4) within a recombinant hybrid pedigree, (5) in the genotype-phenotype map, and (6) in gene expression. Thus, our analyses “link” changes in genotype to patterns of evolution. Finally, we note that we have been very cautious and deliberate in our writing and wording to not describe any of our genomic regions, or specific genes, as causative – as it is likely that a whole host of genetic processes are involved in determining jaw associations, and determining causation will require many additional years of work.

I did not get the part with the F_{ST} threshold of 0.57 (page 19). There is certainly no such thing as a common F_{ST} threshold for cichlids.

Response: This is a good point, and we thank the reviewer for indicating the need for greater clarity. The 0.57 F_{ST} threshold was based on a genus level comparison between two closely related species *Labeotropheus fuelleborni* (also used in this study) and *Metriaclima zebra* (Mims MC, et al. Geography disentangles introgression from ancestral polymorphism in Lake Malawi cichlids. *Mol Ecol.* 19, 940-51 (2010)). Given the close relationship between *Tropheops* and these other genera, we felt that applying this threshold was not inappropriate, but the reviewer’s point is well taken. We have now calculated z-transformed F_{ST} values (zF_{ST}) to more comprehensively set thresholds base on our own empirical data.

While our marker of interest is alternatively fixed between our parental taxa (*Labeotropheus fuelleborni* and *Tropheops sp.* “red cheek”) indicating high divergence, we have also plotted threshold levels on our fine map figures (Fig. 4E). We plot thresholds for $Z=1$ and $Z=2$, which correspond to F_{ST} values of ~ 0.60 and ~ 0.90 respectively. An interesting outcome from this approach is recognizing the similarity between our own F_{ST} scores at $z=1$ – 0.60, and the value reported by Mims and colleagues – 0.57.

REVIEWERS' COMMENTS

Reviewer #1 (Remarks to the Author):

The authors test Liem's pharyngeal jaw decoupling hypothesis across 88 cichlid species at the phylogenetic level and also test for integration between the systems at the population level. The authors go on to test for genetic markers responsible for pleiotropy between both systems using a wide array of analyses.

They find strong integration between the jaw systems at the phylogenetic scale and at the population scale and perhaps most importantly identify a gene responsible for pleiotropy across the oral and pharyngeal jaws.

In general, I think this paper is phenomenal. It represents a deep and succinct dive into integration and takes a novel cutting-edge approach to test for the genetic underpinnings of this important process. From a methodological standpoint the methods look pretty clean to me. My only suggestions pertain to placing their findings into a slightly broader context, and adding some relevant citations. Please see my comments below:

Line 94: The authors should cite Burress and Munoz , 2021 either here, or elsewhere in the manuscript, they report similar findings of integration in cichlids using functional metrics.

Lines 402-436: The authors suggest that integration can act as more than just a constraining force for trait evolution and explain how it might be selectively beneficial to maintain integration between different jaws systems. I completely agree with this finding! I think that it is important to build their interpretations into a broader context beyond cichlids in this paragraph as there is now an emerging synthesis in fishes and beyond that integration can and does facilitate rapid and coordinated trait evolution. I think the authors should cite some of the new work here (e.g. Evans et al., 2021; Watanabe et al., 2021; Watanabe et al., 2019).

Literature Cited

Edward D Burress, Martha M Muñoz, Ecological limits on the decoupling of prey capture and processing in fishes, *Integrative and Comparative Biology*, 2021;, icab148, <https://doi.org/10.1093/icb/icab148>

Evans, Kory M., et al. "Integration drives rapid phenotypic evolution in flatfishes." *Proceedings of the National Academy of Sciences* 118.18 (2021).

Watanabe, Akinobu, et al. "Novel neuroanatomical integration and scaling define avian brain shape evolution and development." *eLife* (2021).

Watanabe, Akinobu, et al. "Ecomorphological diversification in squamates from conserved pattern of cranial integration." *Proceedings of the National Academy of Sciences* 116.29 (2019): 14688-14697.

Reviewer #2 (Remarks to the Author):

I appreciate all the hard work the authors put into their revision and for the most part I am very satisfied with their responses. However, a major comment from my initial review still seems to hang over the manuscript and this needs more attention. The issue has to do with the tendency of the authors to refer to morphological outlier species in a plot of species oral jaw shape vs pharyngeal jaw shape as having weaker integration. The term integration can mean somewhat different things depending on context. This paper is about evolutionary integration, which is measured as the correlation in shape between two structures. Therefore it just doesn't make sense to me to describe an outlier in morphospace as being less integrated – integration here is a property of a group of species, not a single species. I have pointed out specific instances where I think this issue is still confounded in the manuscript. My comments should be quite easy to address so I see no further obstacles.

L81. '...more complex pharyngeal musculature...' is not a feature of pharyngognathy. I think the trait you are after here is the muscular sling – direct muscular connection between the neurocranium and the lower pharyngeal jaw. If anything this system is less complex than the mechanisms and muscular organization of non-pharyngognathous acanthomorph fishes.

L82. I also do not think there is any reason to think of these pharyngeal jaws as being more dexterous. The joint between the neurocranium and upper PJ, the fused lower PJ and the muscular sling result in a stronger bite, but I do not think anyone ever showed that these jaws have greater dexterity.

L156. I am sorry to keep harping on this point but I still find myself confused with the use of the term 'integration'. In this paragraph you are discussing the pattern of oral jaw vs pharyngeal jaw correlation across 88 species. This measures evolutionary integration. This integration is a property of a lineage. You describe an outlier species in that plot as having weaker (or different) integration. But the terminology

doesn't make sense to me – integration is a property of this lineage of cichlids, not a property of a single species. Developmental integration would be a property of a single species. In this example the outlier species has departed from the rift lake cichlid pattern of integration, at some point experiencing an evolutionary history of the two jaw systems that diverged from that found more broadly in the group. If we measure integration as the correlation between oral and pharyngeal jaws, why would we call an outlier species in that plot 'uncorrelated'?

L184. Here again *Tropheops* is described as having strong integration because it fell close to the overall line.

L196. And again here you compared two species that showed different integration strengths in the macroevolutionary analysis (weak vs strong). It just doesn't make sense that an outlier in that plot be called more weakly integrated. In this context integration is not a property of a single species.

Reviewer #3 (Remarks to the Author):

I think that the authors have done a good job revising their paper, toning-down their language, and clarifying the open questions raised by the referees. I also appreciate the efforts of the authors to include additional species and think that increasing the sample size has made the 'integration' part more convincing, although I still believe that the set of species is not representative (and, as Reviewer #2 also rightly pointed out, Malawi-heavy). I thus feel that the authors should not exclude the possibility of a sampling bias when comparing their results between cichlids from lakes Malawi and Tanganyika. Other than that, and a few additional and more minor remarks (see below), I feel that the manuscript is now acceptable for publication.

L100: Please define LOJ and LPJ when first mentioning the abbreviations.

L122-125: Why would e.g. a molluscivore be considered more specialized than any other invertebrate feeder or an algae grazer? Or, to turn around the question: If the only scale-eater in the data set (*Perissodus*), the molluscivores, and the ovivore(s) [which species is that?] are considered specialists, which species are considered generalists? I am wondering how the overall results on 'integration' would have turned out, had the authors analysed a set of species that is balanced between 'specialists' and 'generalists' [aufwuchs feeders?].

L169: Based on what the Tanganyika cichlids "appear more integrated" than the ones from Malawi? Their r-PLS? I would argue that if the two sets of species do not differ significantly, none of them "appears" more or less integrated.

Supplementary Table 1: In this table, some sampling localities are missing, and for a substantial number of specimens there is still no morphosource link (perhaps these are the newly added species). This would need to be fixed.

Also, if the Lake Malawi cichlids are divided into 'Mbuna' and 'Non-mbuna', then at least the tribe (clade) names should be provided for the Tanganyikan representatives as well – especially, because there is no such thing as a Tanganyika clade (the assemblage is not monophyletic). The column 'clade' is in any case misleading, as the clade 'mbuna' is nested within Haplochromini and this nested within the African cichlids (Pseudocrenilabrinae). So all mbuna and non-mbuna are also 'African'. *Orthochromis* should not be listed as 'Tanganyika', as this is a riverine haplochromine cichlid. If this species was included in the PLS comparison in the Tanganyika species set, then I am afraid that the authors will have to rerun this part of the analyses without this taxon (sorry for not having seen this in the first version).

REVIEWERS' COMMENTS

Reviewer #1 (Remarks to the Author):

The authors test Liem's pharyngeal jaw decoupling hypothesis across 88 cichlid species at the phylogenetic level and also test for integration between the systems at the population level. The authors go on to test for genetic markers responsible for pleiotropy between both systems using a wide array of analyses.

They find strong integration between the jaw systems at the phylogenetic scale and at the population scale and perhaps most importantly identify a gene responsible for pleiotropy across the oral and pharyngeal jaws.

In general, I think this paper is phenomenal. It represents a deep and succinct dive into integration and takes a novel cutting-edge approach to test for the genetic underpinnings of this important process. From a methodological standpoint the methods look pretty clean to me. My only suggestions pertain to placing their findings into a slightly broader context, and adding some relevant citations. Please see my comments below:

Thank you for your comments! We're glad you enjoyed the paper.

Line 94: The authors should cite Burress and Munoz , 2021 either here, or elsewhere in the manuscript, they report similar findings of integration in cichlids using functional metrics.
Added.

Lines 402-436: The authors suggest that integration can act as more than just a constraining force for trait evolution and explain how it might be selectively beneficial to maintain integration between different jaws systems. I completely agree with this finding! I think that it is important to build their interpretations into a broader context beyond cichlids in this paragraph as there is now an emerging synthesis in fishes and beyond that integration can and does facilitate rapid and coordinated trait evolution. I think the authors should cite some of the new work here (e.g. Evans et al., 2021; Watanabe et al., 2021; Watanabe et al., 2019).

We have included a section that cites recent work on phenotypic integration and evolutionary constraints on lines 412-414.

Literature Cited

Edward D Burress, Martha M Muñoz, Ecological limits on the decoupling of prey capture and processing in fishes, *Integrative and Comparative Biology*, 2021; icab148, <https://doi.org/10.1093/icb/icab148>

Evans, Kory M., et al. "Integration drives rapid phenotypic evolution in flatfishes." *Proceedings of the National Academy of Sciences* 118.18 (2021).

Watanabe, Akinobu, et al. "Novel neuroanatomical integration and scaling define avian brain shape evolution and development." *eLife* (2021).

Watanabe, Akinobu, et al. "Ecomorphological diversification in squamates from conserved pattern of cranial integration." *Proceedings of the National Academy of Sciences* 116.29 (2019): 14688-14697.

Reviewer #2 (Remarks to the Author):

I appreciate all the hard work the authors put into their revision and for the most part I am very satisfied with their responses. However, a major comment from my initial review still seems to hang over the manuscript and this needs more attention. The issue has to do with the tendency of the authors to refer to morphological outlier species in a plot of species oral jaw shape vs pharyngeal jaw shape as having weaker integration. The term integration can mean somewhat different things depending on context. This paper is about evolutionary integration, which is measured as the correlation in shape between two structures. Therefore it just doesn't make sense to me to describe an outlier in morphospace as being less integrated – integration here is a property of a group of species, not a single species. I have pointed out specific instances where I think this issue is still confounded in the manuscript. My comments should be quite easy to address so I see no further obstacles.

Thank you for your comments.

L81. '...more complex pharyngeal musculature...' is not a feature of pharyngognathy. I think the trait you are after here is the muscular sling – direct muscular connection between the neurocranium and the lower pharyngeal jaw. If anything this system is less complex than the mechanisms and muscular organization of non-pharyngognathous acanthomorph fishes.

We were referring to the muscular sling. We are now more explicit in describing the muscular anatomy of the lower pharyngeal jaw.

L82. I also do not think there is any reason to think of these pharyngeal jaws as being more dexterous. The joint between the neurocranium and upper PJ, the fused lower PJ and the muscular sling result in a stronger bite, but I do not think anyone ever showed that these jaws have greater dexterity.

Changed to reference the stronger bite force.

L156. I am sorry to keep harping on this point but I still find myself confused with the use of the term 'integration'. In this paragraph you are discussing the pattern of oral jaw vs pharyngeal jaw correlation across 88 species. This measures evolutionary integration. This integration is a property of a lineage. You describe an outlier species in that plot as having weaker (or different) integration. But the terminology doesn't make sense to me – integration is a property of this lineage of cichlids, not a property of a single species. Developmental integration would be a property of a single species. In this example the outlier species has departed from the rift lake cichlid pattern of integration, at some point experiencing an evolutionary history of the two jaw systems that diverged from that found more broadly in the group. If we measure integration as the correlation between oral and pharyngeal jaws, why would we call an outlier species in that plot 'uncorrelated'?

Integration operates at multiple different levels: the macroevolutionary level (i.e., among species), the microevolutionary level (i.e., within species), and the developmental level (i.e., individual). We examine some of these levels, but not others, and we think this is where some of the confusion has crept in.

For example, you state, “Developmental integration would be a property of a single species”, but this is not the case. Developmental integration operates at the individual level, and we do not investigate this level of integration in the current study. Measuring developmental integration would involve a correlation between structures within an individual over time. Similarly, this statement “But the terminology doesn’t make sense to me – integration is a property of this lineage of cichlids, not a property of a single species” also confuses part of our central premise, that integration operates *independently* at multiple levels – the strength of integration measured at each level does not need to align. For example, the strength of developmental integration may not have any bearing on the strength of integration at the macroevolutionary level. The same may be true for integration at macro- vs microevolutionary levels, which is precisely why we assess integration at multiple levels.

We have more carefully defined each level of integration that we examine, highlight differences among them, and explain which are assessed in our study. See lines 92-112.

L184. Here again *Tropheops* is described as having strong integration because it fell close to the overall line.

Yes. We explicitly examine integration at this level – the microevolutionary level – in that we examine differences in integration within both *Labeotropheus* and *Tropheops*, a comparison that you note in a comment below (Line 196). We focused on these two taxa, in part, because they showed different patterns in the macroevolutionary population, where one was relatively far from the line (*Labeotropheus*) and one was close (*Tropheops*). Because integration at different levels need not align, we followed this up by assessing integration within each species, and in this case the patterns held, with *Labeotropheus* showing more scatter around the line, and *Tropheops* showing a distribution closer to the line. But our assessment here goes beyond simply, ‘which is closer/further from the line?’ We also use a Z-score comparison test so we could definitively say which taxa was more integrated, and found that that *Tropheops* was more integrated than *Labeotropheus* (at the microevolutionary level).

L196. And again here you compared two species that showed different integration strengths in the macroevolutionary analysis (weak vs strong). It just doesn’t make sense that an outlier in that plot be called more weakly integrated. In this context integration is not a property of a single species.

The comparison here does not hinge on whether these two taxa were far or close to the line of best fit. We empirically tested for differences in integration between the groups by statistically comparing their effects sizes and found evidence for stronger integration in *Tropheops* relative to *Labeotropheus*. See lines 205-209.

Reviewer #3 (Remarks to the Author):

I think that the authors have done a good job revising their paper, toning-down their language, and clarifying the open questions raised by the referees. I also appreciate the efforts of the authors to include additional species and think that increasing the sample size has made the 'integration' part more convincing, although I still believe that the set of

species is not representative (and, as Reviewer #2 also rightly pointed out, Malawi-heavy). I thus feel that the authors should not exclude the possibility of a sampling bias when comparing their results between cichlids from lakes Malawi and Tanganyika. Other than that, and a few additional and more minor remarks (see below), I feel that the manuscript is now acceptable for publication.

Thank you for your detailed comments.

L100: Please define LOJ and LPJ when first mentioning the abbreviations.

Done.

L122-125: Why would e.g. a molluscivore be considered more specialized than any other invertebrate feeder or an algae grazer? Or, to turn around the question: If the only scale-eater in the data set (Perissodus), the molluscivores, and the ovivore(s) [which species is that?] are considered specialists, which species are considered generalists? I am wondering how the overall results on 'integration' would have turned out, had the authors analysed a set of species that is balanced between 'specialists' and 'generalists' [aufwuchs feeders?].

While we appreciate these comments, we were careful not to categorize taxa into specialized vs. generalist bins, as this can be a point of contention for many. Indeed, in a previous submission a reviewer cautioned against this. In response, we were careful to discuss differences among taxa only in terms of feeding morphology, not specialized ecologies.

L169: Based on what the Tanganyika cichlids "appear more integrated" than the ones from Malawi? Their r-PLS? I would argue that if the two sets of species do not differ significantly, none of them "appears" more or less integrated.

We have toned down the language in this section to more clearly state that differences are minimal between taxa in each lake. We simply wanted to note that Tanganyika cichlids had greater Z-scores (indicating stronger integration), which we find notable given their high morphological disparity and positions in morphospace relative to the Malawi members.

Supplementary Table 1: In this table, some sampling localities are missing, and for a substantial number of specimens there is still no morphosource link (perhaps these are the newly added species). This would need to be fixed.

In many cases Morphosource, or the institution that scanned the specimen, did not list locality information. Additionally, in scenarios where we got fish from importers, sometimes they would not list the exact localities. Many of the specimens included in this study were collected and scanned by the authors – in this case there would be no morphosource link.

Also, if the Lake Malawi cichlids are divided into 'Mbuna' and 'Non-mbuna', then at least the tribe (clade) names should be provided for the Tanganyikan representatives as well – especially, because there is no such thing as a Tanganyika clade (the assemblage is not monophyletic). The column 'clade' is in any case misleading, as the clade 'mbuna' is nested within Haplochromini and this nested within the African cichlids (Pseudocrenilabrinae). So all mbuna and non-mbuna are also 'African'. Orthochromis should not be listed as 'Tanganyika', as this is a riverine haplochromine cichlid. If this species was included in the

PLS comparison in the Tanganyika species set, then I am afraid that the authors will have to rerun this part of the analyses without this taxon (sorry for not having seen this in the first version).

We agree that the clade designation is misleading. As you can see from the Figures, we switched this to Lake Basin / Region, but did not update this in the supplementary table. This is now fixed.